# Advances in Cellulose-Based Composites for Energy Applications

**DOI:** 10.3390/ma16103856

**Published:** 2023-05-20

**Authors:** Choon Peng Teng, Ming Yan Tan, Jessica Pei Wen Toh, Qi Feng Lim, Xiaobai Wang, Daniel Ponsford, Esther Marie JieRong Lin, Warintorn Thitsartarn, Si Yin Tee

**Affiliations:** 1Institute of Materials Research and Engineering (IMRE), Agency for Science, Technology and Research (A*STAR), 2 Fusionopolis Way, Innovis #08-03, Singapore 138634, Singapore; tengcp@imre.a-star.edu.sg (C.P.T.); tanmy@imre.a-star.edu.sg (M.Y.T.); jessica_toh@imre.a-star.edu.sg (J.P.W.T.); limqf@imre.a-star.edu.sg (Q.F.L.); wangxb@imre.a-star.edu.sg (X.W.); daniel.ponsford.19@ucl.ac.uk (D.P.); linem@imre.a-star.edu.sg (E.M.J.L.); thitsartarnw@imre.a-star.edu.sg (W.T.); 2Department of Chemistry, University College London, London WC1H 0AJ, UK; 3Institute for Materials Discovery, University College London, London WC1E 7JE, UK

**Keywords:** cellulose, cellulose-based composites, flexible electronics, energy conversion, energy storage, batteries, green energy harvesting

## Abstract

The various forms of cellulose-based materials possess high mechanical and thermal stabilities, as well as three-dimensional open network structures with high aspect ratios capable of incorporating other materials to produce composites for a wide range of applications. Being the most prevalent natural biopolymer on the Earth, cellulose has been used as a renewable replacement for many plastic and metal substrates, in order to diminish pollutant residues in the environment. As a result, the design and development of green technological applications of cellulose and its derivatives has become a key principle of ecological sustainability. Recently, cellulose-based mesoporous structures, flexible thin films, fibers, and three-dimensional networks have been developed for use as substrates in which conductive materials can be loaded for a wide range of energy conversion and energy conservation applications. The present article provides an overview of the recent advancements in the preparation of cellulose-based composites synthesized by combining metal/semiconductor nanoparticles, organic polymers, and metal-organic frameworks with cellulose. To begin, a brief review of cellulosic materials is given, with emphasis on their properties and processing methods. Further sections focus on the integration of cellulose-based flexible substrates or three-dimensional structures into energy conversion devices, such as photovoltaic solar cells, triboelectric generators, piezoelectric generators, thermoelectric generators, as well as sensors. The review also highlights the uses of cellulose-based composites in the separators, electrolytes, binders, and electrodes of energy conservation devices such as lithium-ion batteries. Moreover, the use of cellulose-based electrodes in water splitting for hydrogen generation is discussed. In the final section, we propose the underlying challenges and outlook for the field of cellulose-based composite materials.

## 1. Introduction

Cellulose, one of the most ubiquitous natural biopolymers on Earth, is a sustainable green resource with good versatility, degradability, and biocompatibility [1,2,3]. It can be extracted from a wide range of plants, certain bacteria species, various algae, and specific animals. Among these, plants such as wood, rice husk, sisal, hemp, flax, kenaf, coconut husk, and cotton are the major sources of cellulose, as it is the main component of structural cell walls in plant cells [4]. Particularly, wood and cotton are the primary sources of plant cellulose. Wood, being readily available, is a good starting material for harvesting cellulose. It is a natural composite material with a hierarchical architecture comprising around 40–50% cellulose, with the remaining half consisting of hemicelluloses and lignin [5]. Cotton is also a high-quality source for pure cellulose as it has more than 90% cellulose content [6]. Cellulose is synthesized by bacteria from *Gluconacetobacter*, *Azotobacter*, *Rhizobium*, *Pseudomonas*, *Alcaligenes*, and *Sarcina* species and termed bacterial cellulose [7]. *Gluconacetobacter xylinus* (formerly named *Acetobacter xylinum* and *Acetobacter xylinus*) is most widely used for the production of bacterial cellulose due to its high conversion efficiency and wide acceptance of different carbon sources [8]. Bacterial cellulose is produced in the form of cellulose microfibrils when these bacteria species metabolize carbon- and nitrogen-rich sources in an oxidative fermentation process [9]. The microfibril formation and crystallization of bacterial cellulose can be controlled by modifying the culturing conditions. The differentiating feature of cellulose produced from bacteria compared to most plants is that bacterial cellulose is of high purity, without the presence of hemicelluloses and lignin. Green, red, gray, and brown species of algae are also good sources of cellulose. The algae cell wall contains celluloses which can be extracted through acid hydrolysis and mechanical refining to obtain cellulose microfibrils with high crystallinity and aspect ratios greater than 40 [10]. The purity of algal cellulose varies depending on the algae species and is much lower compared to bacterial cellulose due to hemicellulose and lignin content [11,12]. The only known animals capable of cellulose production are tunicates, which are marine invertebrate animals belonging to the subphylum Tunicata. Tunicin is the animal cellulose component that can be obtained from the tunic exoskeleton of tunicates. The tunicate cellulose, in the form of nano-/micro- fibrils, is highly crystalline with a high aspect ratio around 60 to 70, and a large specific surface area of approximately 150–170 m^2^ g^−1^ [10].

Cellulose is a linear chain, high molecular weight homo-biopolymer, consisting of hundreds to thousands of repeating D-glucose monomers linked by β-1,4-glycosidic bonds [13]. As there are many hydroxyl groups present in the cellulose chain, van der Waals forces and hydrogen bonds are formed intramolecularly and intermolecularly, resulting in parallel assembly of the cellulose chains into elementary fibrils of widths around 3–5 nm and lengths of hundreds of nanometers. Several of these elementary fibrils bind together to form microfibrils of widths around 5–20 nm and lengths of few micrometers [14].

There are four polymorph types of cellulose, namely, celluloses I, II, III, and IV, which are differentiated by their hydrogen bonding sites between and within strands [11,15]. Cellulose I, also known as natural cellulose, has the highest crystallinity with two co-existing crystal phases, cellulose Iα and cellulose Iβ. Cellulose Iα phases are synthesized mainly by bacteria and algae while Iβ cellulose is generally produced in higher plants. Cellulose II is obtained through the treatment of cellulose I with sodium hydroxide, or during the regeneration of cellulose I. Cellulose III is formed during the liquid ammonia treatment of cellulose I/II, while cellulose IV is derived from the heat treatment of cellulose III in glycerol [16]. Cellulose I has the highest elastic modulus value (138 GPa), while celluloses II, III, and IV have lower elastic modulus values (75 to 88 GPa), which are comparable to glass fibers (73 GPa) [17]. The elastic moduli of the various cellulose types differ from one another due to the crystallization transitions taking place during the different treatments, which result in changes in the cellulose backbone conformations and positions of intramolecular hydrogen bonds [18].

Natural cellulose has a unique multi-level architecture consisting of highly ordered crystalline and disordered amorphous regions which can be extracted into nanocrystals, nanofibrils, or other nanostructures through mechanical and chemical approaches, or combinations of both approaches [10]. Cellulose nanocrystals are highly crystalline spindle-like nanoparticles obtained by removal of the amorphous domains, while cellulose nanofibrils are nanoscale fibers with both crystalline and amorphous regions presented in an interconnected network. Other examples of nanocellulose include amorphous nanocellulose and hairy cellulose nanocrystals.

Cellulose has been used for several centuries as wood for fuel and building materials, as fibers to make paper and textile materials for apparels, ropes, etc. As a green and renewable material, cellulose can replace plastic and metal substrates in a broad range of applications to reduce over-reliance on finite and rapidly depleting non-renewable resources and lower negative impact on the environment [2,19,20]. Subsequently, most studies on cellulose have been focused on cellulose-to-fuel conversion, while recently there has been emerging interest in the functionalization and engineering of cellulose-based materials for flexible electronics and green energy conversion systems. Cellulose, with its distinctive structural and chemical characteristics, can be arranged in different nanoscale configurations to fulfil the design requirements of various energy devices [21]. In order to address the up-and-coming applications of cellulose-based materials in energy-related areas, we present the current advancements of cellulose processing in this review, as well as the applications of cellulose in the advancing field of flexible wearable electronics, thermoelectric nanogenerators, mechanical energy nanogenerators, sensors, electrodes, and photovoltaic solar cells (Figure 1).

## 2. Classification of Cellulose and Processing

Nanocellulose is defined as nano-scale cellulose extracted from native cellulose, usually via chemical and/or physical modifications. Nanocellulose can be categorized into the following types: cellulose nanocrystals, cellulose nanofibrils, and hairy nanocrystals (Figure 2).

### 2.1. Cellulose Nanocrystals

Cellulose nanocrystals, the most coveted form of nanocellulose, are crystalline particles in the shape of cylindrical rods with diameters smaller than 10 nm and lengths ranging from 100 to 500 nm. Cellulose nanocrystals are conventionally obtained through the acid hydrolysis treatment of cellulose fibers. Sulfuric acid is typically used to swell the amorphous regions and retain the crystalline parts of the cellulose [22]. The negatively charged sulfate groups on the surface of cellulose nanocrystals produced by sulfuric acid hydrolysis improve the stability of colloidal suspensions but reduce the thermal stability and modify the reactivity of the nanocrystals [23,24,25,26]. The cellulose nanocrystal yield is dependent on acid concentration, hydrolysis temperature, and hydrolysis time. Generally, the yields from acid hydrolysis approach 30%, but yields of 70% have also been reported [27].

Alternatively, cellulose nanocrystals can also be isolated via enzymatic hydrolysis. The nanocrystals obtained from enzymatic hydrolysis have higher aspect ratios and greater thermal stabilities, but lower colloidal stability due to the higher agglomeration tendency in comparison with cellulose nanocrystals produced from acid hydrolysis [26]. Furthermore, this method requires the use of complex enzyme systems, as there are significant difficulties arising from a single enzyme process [28,29].

Some studies have demonstrated the co-production of cellulose nanocrystals together with sugars, through enzymatic hydrolysis in the cellulosic ethanol production process. However, there are limitations associated with the raw materials (e.g., eucalyptus and kraft pulp) used in these studies as they differ from those (e.g., sugarcane bagasse) currently used in the industrial production of cellulosic ethanol [30,31,32].

### 2.2. Cellulose Nanofibrils

Cellulose nanofibrils consist of nano-scale fibers (20–100 nm in diameter and several micrometers in length) entangled together in a network. They comprise regions with amorphous and crystalline domains and are of lower crystallinity compared to cellulose nanocrystals. The production of cellulose nanofibrils from lignocellulosic biomass is achieved through a variety of chemical, mechanical, and enzymatic treatments, or a combination of the methods [33].

TEMPO (2,2,6,6-tetramethylpiperidine-1-oxyl)-mediated oxidation is the most commonly used chemical method to produce cellulose nanofibrils by the introduction of carboxyl groups specifically at C6 of the repeating glucose units [34]. Cellulose nanofibrils produced by this method are around 800 nm in length. However, the toxicity of the catalyst employed in the TEMPO reaction restricts the use of TEMPO-produced cellulose nanofibrils in certain applications [35].

Cellulose nanofibrils can also be obtained by the enzymatic pretreatment of wood pulp, which is a green and low-cost method. Endoglucanase enzymes are used in the hydrolysis process to break down the cellulose into nanofibrils, though this results in cellulose nanofibrils of varied sizes and high average fibril widths [35]. Another frequently used method is ultrafine grinding, in which the cellulose is pulverized into nano-sized fibrils by mechanical shearing forces.

### 2.3. Hairy Cellulose Nanocrystals

Hairy cellulose nanocrystals belong to a newly emerging category of nanocellulose and have a crystalline body with hair-like amorphous cellulose chain protrusions at the two ends. Hairy cellulose nanocrystals are produced by first subjecting cellulose fibrils to periodate oxidation to obtain dialdehyde modified cellulose, followed by heating and/or chemical treatment (chlorite oxidation or Schiff base reaction) to solubilize and selectively cleave the amorphous regions. Depending on the treatment, electrically neutral, negatively or positively charged hairy cellulose nanocrystals can be synthesized [36]. Neutral hairy cellulose nanocrystals, also known as sterically stabilized nanocrystalline cellulose, are obtained by heating dialdehyde modified cellulose fibrils to 80 °C. Negatively charged hairy cellulose nanocrystals (electrosterically stabilized nanocrystalline cellulose) are obtained by reacting dialdehyde modified cellulose fibrils with chlorite at room temperature while positively charged hairy cellulose nanocrystals (cationic- electrosterically stabilized nanocrystalline cellulose) are produced when dialdehyde-modified cellulose fibrils are reacted with Girard’s reagent T at room temperature and pH ~4.5 followed by heating at 60 °C.

Anionic carboxylated hairy cellulose nanocrystals can also be produced via a mechanochemical process from wood kraft pulp through cryogrinding or in combination with mono-chloroacetate under alkaline conditions [37]. Type I allomorphs of cellulose are obtained though cryogrinding while Type II cellulose is obtained through cryogrinding with mono-chloroacetate.

## 3. Cellulose-Based Composites for Flexible and Wearable Electronics

To date, the majority of materials used for flexible electronic devices have been derived from non-renewable synthetic polymers such as poly(methyl methacrylate) [38], polyethylene terephthalate [39], polydimethylsiloxane (PDMS) [40], polyurethane [41], or polymeric network-ionic liquid hybrids [42]. Although these studies show promising results for flexible electronics, the environmental impact of using petroleum-based materials remains a concern. For greener and biocompatible flexible electronics, cellulose has become a favorable fibrous material not only for scientific exploration, but also for practical needs. Several research works have reported using cellulosic materials to develop flexible electronics [43], electronic textiles [44], and wearable sensors [45]. Below, some of the applications of fibrous cellulose, as well as other cellulose structures, for electronic devices are discussed.

### 3.1. Cellulose-Based Materials in Flexible Electronics

Functional materials, such as metal oxides, carbon-based materials, and conductive polymers can be incorporated into cellulosic materials for fabricating conductive electrodes and dielectric layers in flexible electronics [46,47]. Among these materials, silver nanowires are widely used because of their outstanding electrical and optical properties, in addition to their ductility [48]. However, highly humid conditions and weak interfacial interactions between silver nanowires and substrates typically reduce the long-term stability of such conductive networks. Wang et al. [49] designed an assembly/encapsulation integration approach, producing a regenerated cellulose and poly(3,4-ethylene dioxythiophene)-poly(styrene-sulfonate) (PEDOT:PSS) hybrid film with robust interfacial structure to protect silver nanowires from oxidation and maintain both high conductivity (low sheet resistance of 6.9 Ω sq^−1^) and good flexibility. Their work demonstrated the potential of the hybrid film to act as a flexible strain sensor in a high humidity environment. In another work, Park et al. [50] directly constructed silver nanowire-based microelectrodes on regenerated cellulose substrates via a photolithography process, which displayed good adhesion properties. The silver nanowires-patterned regenerated cellulose films were converted into conductive hydrogel films using an aqueous swelling process. These films exhibited high optical transparency and good electrical conductivity, along with high mechanical stability and bendability.

Another approach to develop flexible and degradable electronics is to deposit the active materials on paper [51]. Highly conductive paper can be fabricated through soaking common printing paper in pyrrole monomer solution, followed by polymerization. Porous and flexible conductive paper prepared in this way displayed a high electrical conductivity of 15 S cm^−1^ and a low sheet resistance of 4.5 Ω sq^−1^. Similarly, flexible microwave and digital electronics have been constructed on cellulose nanofibril papers [52]. Cellulose paper has the intrinsic advantage of being biodegradable and so can be fully degraded after an extended period of time, freeing up space in electronic waste disposals.

When flexible 2D graphene oxide sheets were incorporated into a continuous network of closely packed nanocrystals, the graphene-based polymer nanocomposites exhibited high mechanical strength along with excellent toughness, due to the synergistic interaction between the two different nanocomponents. The high concentration of surface anionic functional groups introduces many options for the effective bonding of cellulose nanocrystals to primed graphene oxide sheets via noncovalent yet strong ionic interactions and hydrogen bonding. Accordingly, a laminated nanomembrane consisting of rigid cellulose nanocrystal networks conformally wrapped by 2D graphene oxide sheets was fabricated using a spin-assisted layer-by-layer technique. This approach can be utilized to enhance mechanical performance and achieve high optical transparency in the visible range, as well as electrical conductivity properties which can be used in a wide range of flexible electronic applications [53].

### 3.2. Cellulose-Based Textiles with Integrated Electronics

Throughout the history of mankind, textiles have been woven from natural fiber and cellulose. However, since the middle of the twentieth century, synthetic polymeric fibers have become more popular due to their versatile functionalities. In response to global pressure for sustainable alternatives and a growing awareness of eco-friendly products, there is renewed interest in natural fibers and cellulose-based textiles [54]. Smart/electronic textiles are being viewed as a revolutionary instrument to unobtrusively incorporate sensing, energy harvesting, and other functionalities into textiles [55]. Cellulose-based materials have become a promising and common candidate to replace synthetic-based textiles, in order to ensure that next-generation smart textiles remain sustainable.

Christian and co-workers [56] demonstrated an electrically conducting yarn based on regenerated cellulose spun from an ionic liquid and coated with PEDOT:PSS, which was applied by a continuous roll-to-roll method. The resulting yarn exhibited an electrical conductivity of 36 S cm^–1^, which was further improved to 181 S cm^–1^ upon the addition of silver nanowires. The conducting yarn was resilient to repeated deformation and could be machine washed without a loss of conductivity, demonstrating its electrochemical functionality. In another work, an all-weather highly conductive fabric and fire warning sensor were successfully created by a multi-layered deposition process onto cellulose fabrics. The multifunctional superamphiphobic cellulose fabric exhibited high conductivity and excellent flame-retardant properties [44]. It is worth noting that the as-prepared highly conductive fabric and fire warning sensor exhibited different surface resistances, which were 1.1 Ω sq^−1^ and 1 kΩ sq^−1^, respectively. In addition, the highly conductive fabric showed superior electrical stability, even under extreme conditions such as a complex water phase, oil phase and flame exposure, and bending. Jingchun et al. [57] employed an improved in situ polymerization method, whereby conventional lyocell yarns were fabricated into multifunctional yarn electrodes. The polypyrrole-coated lyocell yarn electrodes exhibited excellent electric heating, good flame-retardant properties, and exceptional electrochemical performances. The polypyrrole-coated lyocell yarn fabrics were flame-retardant and could be applied as a wearable electric heater.

In an effort to develop an electronic textile for wearable energy harvesting and self-powered sensing applications, Hu et al. [58] reported strong, biodegradable, and washable cellulose-based conductive macrofibers assembled by wet-stretching and wet-twisting bacterial cellulose hydrogel integrated with carbon nanotubes and polypyrrole (BC/CNT/PPy) (Figure 3a). The cellulose-based conductive macrofibers presented a high tensile strength of 449 MPa and a good electrical conductivity of 5.32 S cm^−1^, and were simultaneously able to maintain structural stability in water. In addition, a degradation experiment revealed that the cellulose-based macrofibers could be degraded within 108 h in a cellulase solution.

### 3.3. Cellulose-Based Materials in Flexible Sensors

Flexible sensing devices are extensively used in numerous sensing platforms because of the need for human–machine interaction, localized healthcare monitoring, and wearable electronic devices [61,62,63,64]. As opposed to synthetic polymers, cellulose-based sensors offer multiple advantages, including biodegradability, biocompatibility, and low cellular toxicity for uses in medical applications. Giandrin et al. reported a printed electrical gas sensor derived from cellulose fiber-based paper to detect water-soluble gas levels in the atmosphere. This technology harnessed the intrinsic hygroscopic properties of the cellulose fibers within the paper, allowing the use of wet chemical methods for sensing.

Yan et al. [65] constructed 3D macroporous nanopapers consisting of graphene and nanocellulose embedded in an elastomer matrix to fabricate piezoresistive strain sensors. The free-standing flexible nanopapers were prepared by vacuum filtration and their 3D microporous structure facilitated successful embedding into the elastomer matrix to obtain stretchable nanopapers. The hybrid nanopaper could be stretched up to 100%, whereas the graphene-nanocellulose based nanopaper reached a limit of 6% strain at maximum. This graphene-nanocellulose nanopaper (with PDMS matrix) was also implanted on data gloves as a prototype for detecting minute human motions such as finger movements. Likewise, a highly flexible and anisotropic strain sensor has been fabricated based on carbonized crepe paper [59]. Making good use of the distinctive anisotropic structure of the carbonized crepe paper (having both aligned carbon fibers and a corrugated surface), the obtained strain sensor displayed highly anisotropic gauge factors in the tensile bending tests perpendicular and parallel to the direction of the fibers. The carbonized crepe paper-based strain sensors have been used to detect complex human motions and construct a controllable 2-degree-of-freedom machine (Figure 3b), suggesting their potential for applications related to multi-dimensional wearable electronics and smart robots.

Safari and van de Ven developed nanocomposites based on cellulose nanocrystals or anionic hairy cellulose nanocrystals (electrosterically stabilized nanocrystalline cellulose) with carbon nanotubes as humidity sensitive devices [66]. Carbon nanotubes-electrosterically stabilized nanocrystalline cellulose-based composites behave as effective insulators when relative humidity is below 75% with a low direct current conductivity of ~5 µS cm^−1^ which is a few orders of magnitude lower than that of carbon nanotube-cellulose nanocrystals nanocomposites that have a direct current conductivity ~10 mS cm^−1^. However, when relative humidity is above 75%, carbon nanotubes-electrosterically stabilized nanocrystalline cellulose composites behave as conductors with a direct current conductivity of ~1 S cm^−1^, which is an increase of several orders. The huge jump in conductivity when relative humidity is higher than 75% is triggered by breakage of hydrogen bonds of the amorphous chains on the anionic hairy cellulose nanocrystals when there is an substantial increase in moisture uptake from <1.5% to >8% at high relative humidity [66]. Further research on carbon nanotubes-electrosterically stabilized nanocrystalline cellulose nanocomposites may support new advances as humidity sensitive devices.

Li et al. used a dip-coating technique to prepare a wearable strain/pressure sensor based on non-woven fabrics sequentially coated with thermoplastic polyurethane and MXene/cellulose nanocrystals (Figure 3c) [60]. These sensors exhibited stable electrical properties coupled with good sensing properties, due to the hydrogen bonding of the components of the non-woven fabrics and the MXene/cellulose nanocrystal layers which resulted in effective interfacial bonding in the commensurable structure. As a result of the stiff conductive layer and stretchy thermoplastic polyurethane matrix, a sensor with a distinctive micro-cracks mechanism was obtained. The conductive non-woven fabrics were subjected to a pre-stretching treatment coupled with stacking of the conductive (modified cellulose nanocrystal) layer, and adjustment of the crack density allowed for tunable strain detection. The strain sensor exhibited a wide sensing range with value of 83%, high sensitivity (gauge factor = 3405), and a low detection limit of 0.1%. A variety of applications including human motion detection, signal collection, and health monitoring in stretchable e-skins and other wearable electronic devices have been realized due to the excellent performance of these sensors.

Apart from cellulose nanocrystals, conductive hydrogels have also shown increasing promise in the field of flexible strain sensors [67,68]. However, their application is greatly restricted due to their low conductivity and poor mechanical properties at subzero temperatures. Chen et al. [69] proposed a method to develop a conductive hydrogel with concurrent antifreezing properties and desirable mechanical properties, for usage in wearable strain sensors. The stretchable and conductive cellulose hydrogel was constructed by grafting acrylonitrile and acrylamide copolymers onto cellulose chains in the presence of zinc chloride, using ceric ammonium nitrate as the initiator. The hybrid hydrogel displayed stretchability (1730%), good tensile strength (160 kPa), high elasticity (90%), good toughness (1074.7 kJ m^−3^), and fatigue resistance, due to the presence of dipole-dipole interactions and multiple hydrogen-bonding interactions within the hydrogel network. Furthermore, the incorporation of zinc chloride resulted in a cellulose-based hydrogel with exceptional electric conductivity (1.54 S m^−1^) and superb antifreezing performance (−33 °C). Finally, the hydrogel showed the high sensitivity and stability required to successfully monitor human activities.

Han et al. [70] prepared an organic mixed ion-electron conducting aerogel to create a single sensor with multiple sensing functionalities. The aerogel was fabricated by freeze drying a water dispersion involving four organic components: poly(3,4-ethylenedioxythiophene) for its electrically conductive characteristics to supply the electronic thermovoltage, poly(styrene sulfonate) as the ionic conducting polymer to give the ionic thermos voltage peak, nanofibrillated cellulose to provide the mechanical integrity of the aerogel, and glycidoxypropyl trimethoxysilane to function as crosslinker and introduce elasticity. The aerogel could act as a multiparameter (pressure, temperature, humidity) sensor and these parameters have significant relevance to various applications in medical diagnostics, food monitoring, safety, and security.

There are practical needs of a sensing system based on thin functional fibers. In particular, there is a requirement for simplicity and an immersive user experience for a human–machine interface. Thus, Guan et al. demonstrated a hierarchically porous, conductive thin fiber based on silver nanowire-bacterial cellulose coated with PDMS dielectric elastomers that can be applied as a flexible sensor for detecting both the pressure and proximity signals of human fingers [62]. The conductive fiber, with a diameter of 53 μm, was synthesized via continuous wet spinning. The wet-spun fiber displayed a tensile strength of 198 MPa with elongation at break of 3.0%, as well as electrical conductivity of 1.3 × 10^4^ S cm^−1^. The structure of the produced fiber generated a large capacitive change during contact mode and induced a good sensitivity of 5.49 kPa^−1^ with a large detection range up to 460 kPa. In touchless mode, the sensor could detect objects up to 30 cm away.

## 4. Cellulose-Based Composites for Energy Conversion

Renewable resources are produced in abundance on Earth, and include hydropower [71,72], wind power [73], and solar power [74,75]. However, harvesting these energies often requires large and complex infrastructures, which make it challenging to harness renewable energy sources in miniaturized and portable electronics. Harvesting energy from non-conventional sources in the environment presents an alternative for low-power applications. Although the energy harvested is small (in the order of milliwatts), enough power can be provided to drive wearable electronic devices, sensor networks in the Internet of Things, and other low-power applications. Thus, the emerging direction of research lies in the development of nanogenerators that can collect and convert micromechanical energy or low-grade thermal energy into electricity efficiently. As discussed in the previous section, cellulose-based materials can be incorporated as components in flexible electronic devices. Likewise, cellulose-based materials have the potential to provide the energy needed for low-power devices by means of conversion. Cellulose-based composites that show potential for the conversion of mechanical, thermal, and solar energy into electrical energy for low-power applications are discussed in this section.

### 4.1. Mechanical Energy Conversion

Mechanical energy can be easily sourced from the surrounding environment. Examples of such motions are wind current [76,77], human body motion [78,79], ocean waves [80], the movement of wildlife [81], sound [82], and mechanical vibrations [83]. Perhaps the two most promising energy harvesting technologies are piezoelectric nanogenerators (PENGs) and triboelectric nanogenerators (TENGs), which can harvest ambient mechanical vibrations via the piezoelectric or triboelectric effect, respectively.

#### 4.1.1. Cellulose-Based Triboelectric Generator

Triboelectric nanogenerators (TENGs) are a newly discovered type of energy generator coined by the Wang group in 2012 for harvesting small-scale low-frequency mechanical energy [84]. The principle of the TENG is based on the coupling effects of triboelectrification and electrostatic induction, whereby electrostatic charges are formed on the interfaces between two different materials during physical contact. The contact-induced triboelectric charges can generate a potential drop and drive electron flow by a mechanical force [85]. In a typical TENG, there is usually a pair of triboelectric materials with different charge affinities that are used to generate charges. The material options for triboelectric layers are diverse, including polymers, metals, and inorganic materials [86]. The most widely used materials are dielectric polymers such as polyimide [87], PDMS [88], polytetrafluoroethylene (PTFE) [89], fluorinated ethylene propylene (FEP) [90], perfluoroalkoxylalkane (PFA) [91], polyvinylidene fluoride (PVDF) [92], etc. The inherent properties (e.g., triboelectric charge density, dielectric properties) of these materials ensure the superior power density of devices. However, these are non-renewable and cause adverse effects to the environment over extended time periods. Thus, it has become critical in recent years to use sustainable materials to fabricate TENGs. Examples of such green materials are PLGA [93], silk [94], chitosan [95], cellulose [96], and starch [97]. Particularly, cellulose-based materials have proven to have excellent biodegradability and biocompatibility in TENG applications, and the performance of TENGs which feature cellulose materials has markedly improved.

As demonstrated by Hu et al. [58], their fabric-based TENGs, which utilized a conductive macrofiber of bacteria cellulose, polypyrrole, and carbon nanotubes as electrodes, showed a maximum open-circuit voltage and short-circuit current of 170 V and 0.8 µA, respectively. With an output power of 352 µW and a degradation time within 108 h, the cellulose-based fiber exhibited the potential to power wearable electronics commercially and environmentally. On the other hand, the excellent performance of a green TENG constructed using cellulose-based tribolayers reported greater than 300 Wm^−2^ of output power density, which remained stable after 30,000 operation cycles in 30–84% humidity. With a high positive charge density, the TENG demonstrated its applicability for use in a broad range of applications [98].

Other than being strong and biodegradable, the practical applications of TENGs require them to be flexible and deformable to ensure favorable accommodation of the TENG to arbitrary surfaces or moving objects. Wang et al. [99] demonstrated a highly stretchable cellulose-based hydrogel electronic device. Despite the inherent strong intermolecular interactions in the cellulose, its chain stiffness could be overcome at 74% of elongation at break, with the incorporation of polyvinyl alcohol. As such, the conductive cellulose/polyvinyl alcohol hydrogel-TENG was able to function well as a pressure, temperature, and strain sensor. Martinez and co-workers [100] fabricated self-powered paper-based electronics via sequential spray-deposition of fluoroalkylated organosilanes, PTFE, ethyl cellulose, and conductive nickel nanoparticles (Figure 4a). By preserving the mechanical properties of the substrate (cellulose paper) upon contact with aqueous and organic liquids, the omniphobic surface of the paper-based electronic device was able to generate power densities up to 300 μW cm^−2^ via harvesting the electrostatic energy of the user. This enabled the use of this device as a wireless battery-free human–machine interface.

Another feasible fabrication technique is to 3D print cellulose based TENGs with nano/micro hierarchically patterned structures (Figure 4b), with voltage output close to 175% of that already achieved by devices obtained from traditional molding methods [101]. The triboelectric response of the all-printed device was significantly enhanced due to the improved surface roughness, contact area, and mechanical resilience which resulted from the nano-porous cellulose nanofiber aerogel structure. The TENGs reported were able to harvest mechanical energy efficiently enough to run 88 light-emitting diodes (LEDs).

Zhang et al. [102] developed an all-cellulose energy harvesting device, which was composed of pure and conductive bacterial cellulose as both the friction layers and electrodes. Due to the widely distributed -OH groups and the porous cellulose network structure, conductive bacterial cellulose could be fabricated by modifying the surface with polypyrrole and carbon nanotubes. With a maximum open-circuit voltage of 29 V, short-circuit current of 0.6 μA, and output power of 3 μW, the sandwich-structured device prepared was capable of driving commercial electronics. Their work demonstrated the feasibility of precise fingertip control by playing a sewable electronic piano (Figure 4c) and “Greedy Snake” cyber game (Figure 4d). In addition, the all-cellulose composites could be degraded completely in 8 h, leaving the remaining carbon nanotubes available for reuse.

**Figure 4 materials-16-03856-f004:**
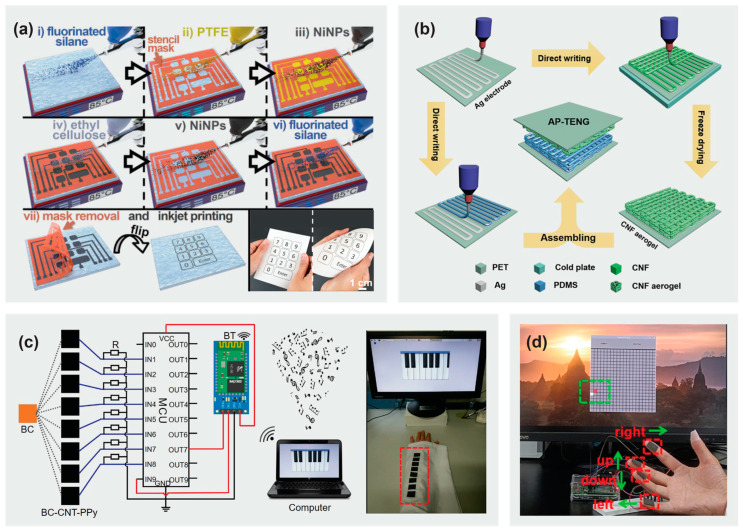
(**a**) Schematic illustration of the fabrication process of a self-powered, paper-based flexible keypad. Reprinted from [100] with permission from Elsevier 2020. (**b**) Schematic illustration of the fabrication process of cellulose nanofiber-based TENG by 3D printing. Reprinted from [101] with permission from Elsevier 2019. Schematic diagram of (**c**) wearable electronic piano and (**d**) cyber game based on all cellulose TENG. Reprinted from [102] with permission from Elsevier 2021.

#### 4.1.2. Cellulose-Based Piezoelectric Generators

The concept of a piezoelectric nanogenerator (PENG) was introduced in 2006 [103] to harvest different types of energy from the surroundings, and Maxwell’s displacement current is the driving mechanism for its operation [104]. The most commonly used piezoelectric materials are ceramics, such as lead zirconate titanate. Although these nanogenerators display high performance, they generally present poor mechanical properties (e.g., brittleness) and toxic composition, which limit their development for biomedical applications and pose challenges for their disposal [105]. Cellulose is a renewable type of piezoelectric polymer and a potential replacement for the ceramic materials used to construct PENGs. Examples of sources for such cellulose-based biopolymers include oak, wood, spruce, etc. [106]. The piezoelectric response is due to the heterogeneous deformation of polar atomic groups from the asymmetric crystalline structures within the material [107]. A characteristic cellulose-based PENG involves three parts, namely, a middle piezoelectric active layer, two conducting electrodes layers, and an exterior packaging layer [105].

Sun et al. [108] reported a wood sponge-based PENG fabricated by a one-step chemical delignification process (Figure 5a). As compared to native wood, an 85-fold performance enhancement with instant voltage of up to 0.69 V and a current of 7.1 nA could be generated with the application of a fairly small stress of 13.3 kPa, due to the increased compressibility of the material. The versatility of the nanogenerator led its applications to range from a wearable movement monitoring system to powering simple electronic devices. Alam et al. [109] reported a PENG composed of native cellulose microfibers and PDMS, with multi-walled carbon nanotubes as conducting filler. The PENG delivered an open circuit voltage of ~30 V and power density of ~9.0 μW cm^−3^ under repeated hand punching, capable of lighting up numerous LEDs or other portable electronic devices, such as a calculator, liquid-crystal display screen, and wristwatch.

To improve the piezoelectric response, the incorporation of inorganic nanoparticles into cellulose has been reported. The high surface-to-volume ratio of nanoparticles and other surface effects plays a significant role compared to corresponding bulk materials, leading to a greater deformation of the piezoelectric nanostructures by comparatively smaller forces [110]. BaTiO_3_ nanoparticles have been extensively studied as one of the non-lead piezoelectric materials with a high piezoelectric coefficient and dielectric constant. With 40 wt% BaTiO_3_ nanoparticles, a nanocellulose composite film attained a maximum piezoelectric output voltage of ~2.86 V, a current of ~262.4 nA, and an electric power of ~378.2 nW under a fairly low compressional stress of 5 kPa [111]. Developed using an aqueous suspension casting technique, the nanocellulose composite film was capable of charging microcapacitors after rectification.

**Figure 5 materials-16-03856-f005:**
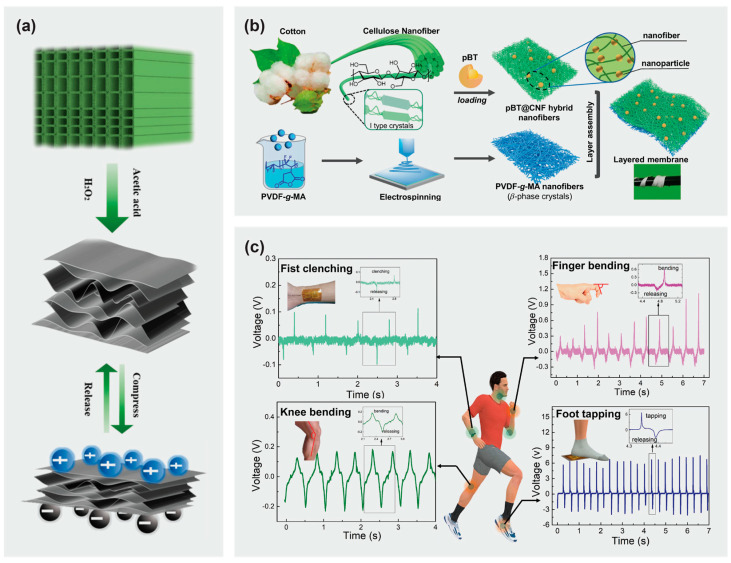
(**a**) Schematic illustration of the fabrication process of wood sponges from native balsa wood and their piezoelectric behavior. Reprinted from [108] with permission from the American Chemical Society 2020. (**b**) Schematic illustration of the fabrication of layered cellulose-based nanofibrous membranes. (**c**) Sensing performance of the cellulose-based nanofibrous membranes for detecting human motions including fist clenching, finger bending, keen bending, and foot tapping during a walking movement. Reprinted from [112] with permission from Elsevier 2022.

In a gold nanoparticles-cellulose/PDMS nanocomposite, an enhancement in the dielectric constant was ascribed to cellulose, and a reduction of dielectric loss was recorded which resulted from the incorporation of gold nanoparticles [113]. Likewise, the piezoelectric output voltage and the current were enhanced due to the presence of gold nanoparticles in the nanocomposite. A mechanical energy harvesting device based on the gold nanoparticle-cellulose/PDMS nanocomposite delivered a superior open circuit voltage of 6 V, a short circuit current of 700 nA, and a peak power density of 8.34 mW m^−2^ without executing any electrical poling steps. The PENG could charge a 10 mF capacitor to 6.3 V and light up two blue LEDs for 677 s concurrently, with a high energy conversion efficiency of 1.8%.

Wang et al. [112] developed PENGs based on layered membranes made of cotton cellulose interfaced maleic-anhydride-grafted PVDF nanofibers. Using a vacuum-assisted layer assembly approach, a hierarchical layered architecture with interlocked interlayer interfaces was fabricated. Core-shell structured polydopamine@BaTiO_3_ nanoparticles were chemically loaded onto the surfaces of cellulose nanofibers, playing the role of an interlayer bridge to bind hydrophilic cotton cellulose and hydrophobic maleic anhydride grafted PVDF layers (Figure 5b). The layered PENGs delivered a maximum piezoelectric coefficient of 27.2 pC N^−1^ and a power density of 1.72 μW cm^−2^. Having excellent mechanical integrity and good electricity generation capabilities, the layered PENG was suitable for usage in self-powered sensors for monitoring human physiological motions such as fist clenching, finger bending, knee bending, and foot tapping (Figure 5c).

### 4.2. Thermoelectric Energy Conversion

Of all primary energy produced globally, more than 72% is lost in the form of heat. For instance, an engine in a car only uses about 30% of the energy from gasoline, while the rest is dissipated as heat. Therefore, to recover even the slightest fraction of that lost energy would make a valuable contribution to combating climate change. One of the possible ways to achieve this is by using thermoelectric materials, which can convert waste heat into useful electricity by harnessing the Seebeck effect. Through a change in temperature across a semiconductor material, voltage can be created for electricity to flow. The performance of thermoelectric materials is evaluated by combining their Seebeck coefficient (*S*), electrical conductivity (*σ*), and thermal conductivity (*κ*) to obtain *ZT*, the dimensionless figure of merit (*S^2^σT/κ*) [114,115,116,117].

With the development of the “Internet of Things”, the persistent demand for self-powered electronics is ever increasing. Thermoelectric generators, which can harvest energy from waste heat, have been extensively researched to address this need [118]. Flexible thermoelectric materials consist of a conductive polymer, conductive filler composites, cellulose (as a flexible substrate), and thermoelectric thin films on flexible substrates [115,119]. A free-standing single-walled carbon nanotube/cellulose acetate composite film has been measured to produce optimal p-type and n-type power factors of 1.41 and 0.516 μW cm^−1^ K^−2^, respectively, at room temperature [120].

Recently, paper-based thermoelectric generators have garnered increasing research interest due to their light weight, flexibility, and ductility which allows them to conformally wrap around heat sources. Paper-based thermoelectric generators can be fabricated by depositing or coating conductive films onto paper substrates [121]. These works have successfully circumvented the brittleness of inorganic semiconductors to accomplish flexible thermoelectric generators and demonstrated their possible applications. For example, a flexible thermoelectric paper was prepared by dipping copy papers into Bi-doped PbTe and PbS solutions, and a power factor of 10 μW mK^−2^ was achieved at 400 K [122]. To improve the power factor, the electrical conductivity of the inorganic thermoelectric films on paper substrates can be increased. Gao et al. [123] prepared a dense paper-based n-type Ag_2_Te nanowires film by transferring the film from glass-fiber sheet to copy-paper using a cold press method. Under a compressive stress, the grain boundaries in the Ag_2_Te nanowires film diminished, leading to an improved electrical conductivity and power factor (192 μW m^−1^ K^−2^ at 195 °C). Jin et al. [124] used a sputtering technique to form Bi_2_Te_3_ films on both sides of a cellulose fiber substrate, and achieved a maximum power factor of 240 μW m^−1^ K^−2^.

Another example of a paper-based thermoelectric generator is a combination of copper iodide- and bismuth-coated cellulose papers, which generated an output voltage of 84.5 mV and corresponding output power of 215 nW at a temperature difference of ~50 °C [125]. Dong et al. [126] modified cellulose paper by introducing doped-Bi_2_Te_3_ and doped-Sb_2_Te_3_ into a paper matrix via a vacuum-assisted filtration process. The resultant paper-based thermoelectric generators, with three units of P–N modules, delivered ~41.2 mV at a temperature difference of 50 K. On the other hand, Li et al. [127] prepared free-standing paper-based thermoelectric generators based on multi-walled carbon nanotubes and carboxylated nanocellulose through a stencil printing method. The findings revealed that higher amounts of carboxyl groups in carboxylated nanocellulose fibers lead to better dispersibility of multi-walled carbon nanotubes and higher thermoelectric performance of the resultant thermoelectric paper. The paper-based thermoelectric generator prototype demonstrated an open circuit voltage of 3.3 mV and an output power of up to 3.7 nW under a temperature gradient of 70 K.

However, there are several challenges faced when applying thermoelectric films on flexible substrates, such as the adhesion and delamination of the film, the differential thermal expansion of the film compared to the substrate, etc. To address these problems, bacterial cellulose with a 3D network can be prepared. Abol-Fotouh et al. [128] fabricated thermoelectric paper by in situ culturing of bacteria in media that contained carbon nanotubes (Figure 6a). As such, the bacterial cellulose was uniquely grown. The resultant freestanding conducting papers could be bent effortlessly, with the bacterial cellulose/carbon nanotube composites being thermally stable up to 200 °C. The grown films may serve as both the active and separating layer and insulate each thermoelectric leg from the adjacent neighbors due to the vertical phase separation of the carbon nanotubes in the bacterial cellulose composite.

Palaporn et al. [119] fabricated a flexible thermoelectric bacterial cellulose/Ag_2_Se nanocomposite paper via the in situ synthesis of Ag_2_Se inside a bacterial cellulose structure. The compacted microstructure enhanced the electrical conductivity to 23,000 S m^−1^ at 400 K, and the Seebeck coefficient to −167 μV K^−1^ at 400 K. A flexible thermoelectric module was assembled from the bacterial cellulose/Ag_2_Se nanocomposite paper, which could harvest up to 12 mV at a temperature difference of 25 K (Figure 6b). A robust honeycomb-like thermoelectric generator was developed by Zhao et al. [129]. The formulated ink, which was composed of Bi_2_Te_3_ and bacterial cellulose, was printed on a paper surface and folded into a honeycomb-like thermoelectric generator (Figure 6c). At a 55 K temperature difference, the thermoelectric generator displayed 96 p–n junctions, attaining a maximum voltage and output power of ~70.5 mV and ~596 nW, respectively (Figure 6d). Furthermore, the honeycomb structure was proven to be highly durable.

**Figure 6 materials-16-03856-f006:**
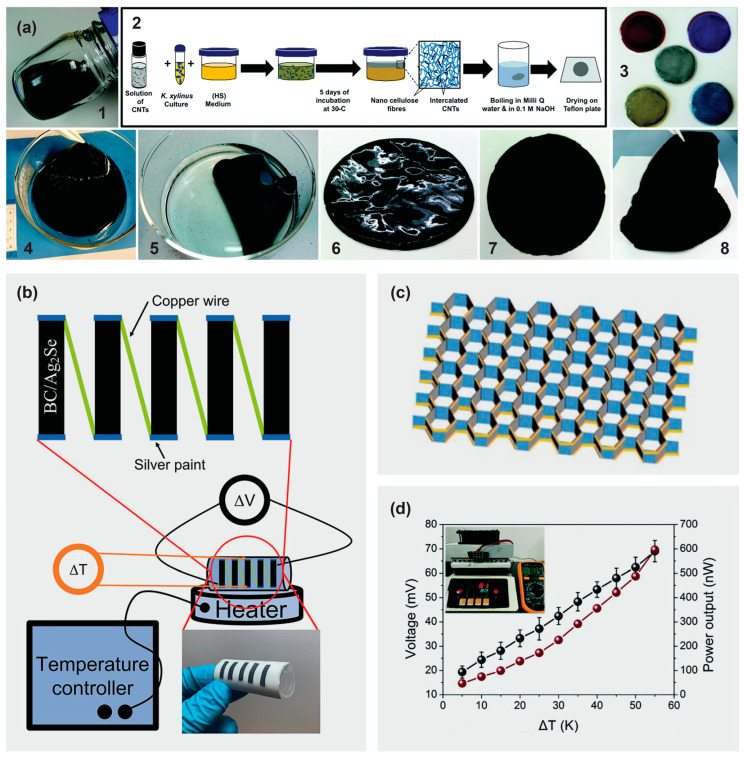
(**a**) Fabrication process of bacterial cellulose/carbon nanotube composites and photographs showing wet and dried film. Reprinted from [128] with permission from the Royal Society of Chemistry 2019. (**b**) Flexible thermoelectric module consisting of five uni-legs made of bacterial cellulose/Ag_2_Se paper interconnected in series. Reprinted from [119] with permission from the American Chemical Society 2022. (**c**) Honeycomb-like paper-based thermoelectric generator with eight sets of honeycomb units. (**d**) Output voltage and power characteristics of the honeycomblike paper-based thermoelectric generator on a planar heat source. Reprinted from [129] with permission from the Royal Society of Chemistry 2019.

### 4.3. Solar Energy Conversion

Solar energy is an inexhaustible resource with vast potential as the cleanest and sustainable energy source for electricity generation [75,130]. While photovoltaic technologies have made great strides in recent years, significant advances are still needed to enable these technologies to be economically and technically viable, as well as environmentally friendly from a life-cycle perspective. Recently, cellulose-based materials have been increasingly applied in many types of optoelectronic devices [131]. Particularly, transparent cellulose nanopaper has emerged as a renewable and eco-friendly material, which displays high optical transmittance due to its nanoporous structure and tunable optical haze [132]. Nanopaper showed no glass transition temperature and thermal degradation up to 300 °C [133]. These papers are attractive substitutes for petrochemical-based substrates for the realization of sustainable solar cell technologies.

An earlier study by Hu et al. [134] demonstrated that transparent and conductive nanocellulose paper derived from cellulose nanofibrils could be produced as a substrate for solar cells with a power conversion efficiency of 0.40%. The nanocellulose paper exhibited large light scattering in the forward direction, which is beneficial for solar cell applications. To make a transparent conductive nanopaper for an optoelectronic device, the nanocellulose paper was deposited with a variety of conductive materials, such as tin-doped indium oxide, carbon nanotubes, and silver nanowires.

A recent study shows that cellulose nanopaper derived from biomass, such as tobacco stalks of agroforestry residues, can be used for light trapping and wide-angle capturing in organic solar cells [135]. The coexistence of inherent crystalline and amorphous regions of cellulose is favorable for the harvesting of oblique incident light to attain a wide-angle absorption, due to the different refractive index of the cellulose itself. The cellulose nanopaper-based device accomplished a power-conversion efficiency of 16.17% and demonstrated an effective wide-angle harvesting from −45 to 45°. The enhanced efficiency and capture angle were ascribed to the combined effect of the light-scattering and the different refractive index of the cellulose itself, along with the effective trapping of light.

Likewise, carboxymethyl cellulose sodium derived from rice straws of agroforestry residues can be applied as a cathode interlayer for organic solar cells in inverted structure [136]. In this device, carboxymethyl cellulose sodium was employed as an interfacial modifier with ZnO for the transfer and collection of electrons. Benefiting from the co-interface layer that can reduce work function, increase absorption, and better interfacial contact, the power conversion efficiency of the device was remarkably improved to 12.01%, which is 9.4% higher than that of pristine ZnO-based organic solar cells. Importantly, studies have shown that cellulose-based solar cells can be separated and recycled into their major components, thus opening the door for a sustainable solar cell recycling industry [137,138].

A cellulose derivative, *O*-(2,3-dihydroxypropyl) cellulose, has been studied as a flexible solar cell substrate [139]. Synthesized by homogeneous etherification, the resultant cellulose-based paper shows a high level of transparency and good ductility, but poor mechanical properties, which is unfavorable in a substrate used for flexible polymer solar cells. To improve the mechanical properties of soft *O*-(2,3-dihydroxypropyl) cellulose, the matrix was reinforced with rigid tunicate cellulose nanocrystals which form strong hydrogen bonding interactions, arising from the surface polyhydroxyl structure. Due to the good interface compatibility between tunicate cellulose nanocrystals and *O*-(2,3-dihydroxypropyl) cellulose, the tensile strengths and toughnesses of nanocomposite papers were improved while retaining high transparency. With tin-doped indium oxide coating on the adhesive transparent paper, the resultant flexible inverted polymer solar cells exhibited a power conversion efficiency of 4.98%.

Apart from the effects on mechanical properties, the treatment of cellulose surfaces is important because cellulose paper is known to be susceptible to water damage and became creased as a result of the swelling of nanocellulose. To solve this problem, the cellulose paper can be coated with acrylic resin to improve on the waterproof property [140]. The acrylic resin used in the case produced a protective coating on the surface of cellulose paper and shielded it from water, thereby leading to enhanced properties of the cellulose paper including transmittance, tensile strength, and thermal stability. Integrating the cellulose paper with a perovskite layer yielded flexible perovskite solar cells with a power conversion efficiency of 4.25% with good stability, maintaining more than 80% of their original efficiency after bending 50 times. Moreover, the cellulose-based paper solar cell was biodegradable and could be disposed of easily by flame burning, showing the great potential of making a renewable energy device without adverse environmental impact.

## 5. Cellulose-Based Composites for Energy Storage

Due to the importance of electronics in our modern society, the global demand for energy has been growing rapidly and there is a need to look into safe, sustainable, and low-cost energy storage devices. The lithium-ion battery has been a keystone in the development of energy storage devices, where it can be found in aerospace technology, portable electronic devices, and electric vehicles [141,142]. Typically, lithium-ion batteries comprise of four main components: the cathode, electrolyte, separator, and anode. To produce greener, safer, and more cost-effective lithium-ion batteries, an alternative source of these components could be cellulose [143]. Cellulose is a highly abundant biopolymer that is versatile enough to be used across the different components of a battery. In this section, we discuss how cellulose can be transformed into porous electrode architectures, followed by the use of cellulose as a separator and solid polymer electrolyte, in a bid to improve battery safety. Lastly, we will delve into the application of cellulose in the next generation of energy storage devices, rechargeable aqueous batteries.

### 5.1. Cellulose-Based Electrodes

Conventionally, graphite is the chosen anode material for lithium-ion batteries due to its good conductivity, thermal stability, and long cycle life. However, there are certain issues with the graphite anode, ranging from incompatibility with other ion batteries to a relatively low capacity (372 mA h g^−1^) [144,145]. As a low cost and sustainable replacement, cellulose can also be pyrolyzed into hard carbons that are compatible with sodium ion batteries [146]. Notably, when freeze-dried and pyrolyzed, cellulose carbon aerogel can form a 3D interconnected network that is also mechanically robust and electrically conductive [147]. This 3D porous architecture can hold active materials and account for their volume changes during battery cycling, thus preserving the structure of the electrodes and prolonging their cycle life [148].

As cellulose-derived carbons also face similar issues such as graphite in fulfilling demands for higher energy density, utilizing silicon [149,150,151], heterodoping [152,153,154], or the addition of transition metal oxides [155,156,157] is necessary for the development of advanced anode materials. The usage of silicon-based anodes such as Li_15_S_4_ is attractive due to their high theoretical capacity of 3579 mA h g^−1^, but is hindered by the poor electrical conductivity of Si [158]. Wang et al. [159] reported a binder-free anode that was able to incorporate an incredibly high amount of silicon of up to 92%. As shown in Figure 7a, this structure involves cellulose nanosheets being rolled up and trapping silicon nanoparticles in a carbon nanotube/cellulose-derived carbon network (Si@CNT/C-microscrolls). A high silicon content of 85% was incorporated and the resultant electrode was able to cycle successfully for 300 cycles at >2000 mA h g^−1^ and volumetric capacity of 930 mA h cm^−3^.

Through nitrogen doping during the pyrolysis of conductive polymer polyaniline crosslinked with bacteria cellulose, Illa et al. [160] developed a fibrous carbon network with a core-shell structure of a bacteria cellulose fibrous carbon core and a polyaniline-derived granular carbon shell. It was able to achieve reversible capacities of 433 and 127 mAh g^−1^ at 1 and 10 °C for 200 cycles, with capacity retention of 99.1 and 74.8%, respectively. The cyclic stability was explained by the well-connected network with large surface area (2037 m^2^ g^−1^) facilitating the uninterrupted lithium-ion/electron transportation.

Another advanced anode material is Fe_3_O_4_ (theoretical capacity of 926 mAh g^−1^). Wang’s group [161] was able to make use of oxidation-aminolysis reactive pyrolysis to convert cellulose paper into graphene-like carbon paper. This graphene-like conductive 3D network has high surface area of 2443 m^2^ g^−1^, which aids in confining the Fe_3_O_4_ nanoparticles to the surface. With the addition of a reduced graphene oxide layer, this further improved the charge transfer flow along the network. This sandwich structure with carbon paper@Fe_3_O_4_@reduced graphene oxide (Figure 7b) was able to cycle stably for >2000 cycles without any obvious capacity fading at 0.5 A g^−1^ (Figure 7c). The stable cycling performance was attributed to the sandwich structure, which was able to accommodate the volume change during the lithiation and delithiation of Fe_3_O_4_. Maintaining anode structural integrity is important to ensure a stable and long cycling life.

Similar to anodes, the cathode materials used for lithium-ion batteries are limited by low capacity (<200 mAh g^−1^), in which the thickness of the cathode coating affects the electrochemical performance of the cell, with suitable thicknesses usually ranging from 50 to 100 µm [162]. Having a thinner cathode coating results in less active material being added, thus resulting in lower capacity. However, cathodes with higher active material loading are too thick, suffer from poor mechanical stability, and have slower ion transport, which also affects the charge transfer throughout the cathode.

To address this problem, Hu’s group [163] developed a thick electrode (~320 µm) with high active material loading of up to 60 mg cm^−2^. The nanopaper electrode comprised a negatively charged cellulose nanofiber network with neutral carbon black nanoparticles absorbed via electrostatic attraction and lithium iron phosphate as the active material. Once the electrode formulation was freeze-dried into aerogel architecture, it could be compressed to less than 5% of its original volume, producing a compact nanopaper electrode. When compared with the conventional lithium iron phosphate electrode which experienced drastic capacity fading from 85.9 to 43.3 mAh g^−1^ after 120 cycles (at 2 mA cm^−2^ and mass loading of 20 mg cm^−2^), this nanopaper electrode was able to cycle up to 150 cycles with 91% capacity retention (initial specific capacity at 133 mAh g^−1^). The nanopaper electrode was able to achieve a real capacity and volumetric energy density of 8.8 mAh cm^−2^ and 538 Wh L^−1^ at a high active material loading of 60 mg cm^−2^ (Figure 7d). Due to the mechanical robustness of the cellulose aerogel, the deformation did not break the electrical connection of the conductive network (Figure 7e), which enabled a high loading of active material to be incorporated without compromising the charge transfer of the cathode.

A 3D conductive cellulose matrix can also be employed in lithium-sulfur batteries, which aids with prevention of loss of active material from the cathode where polysulfide is continuously dissolved in the liquid electrolyte [164]. Recently, porous cellulose-based pyrolyzed carbon aerogels with large surface area have been developed [165,166]. The cellulose-based aerogels with interconnected network are a good anchoring host for sulfur, thus improving the interface between sulfur and electrolyte, while limiting the loss of the lithium polysulfide intermediates into the electrolytes. Moreover, these cellulose-based aerogels can be sourced from cellulose acetate that is extracted from waste cigarette filters, which not only enables upcycling but also reduces the environmental pollution hazard [157,166].

**Figure 7 materials-16-03856-f007:**
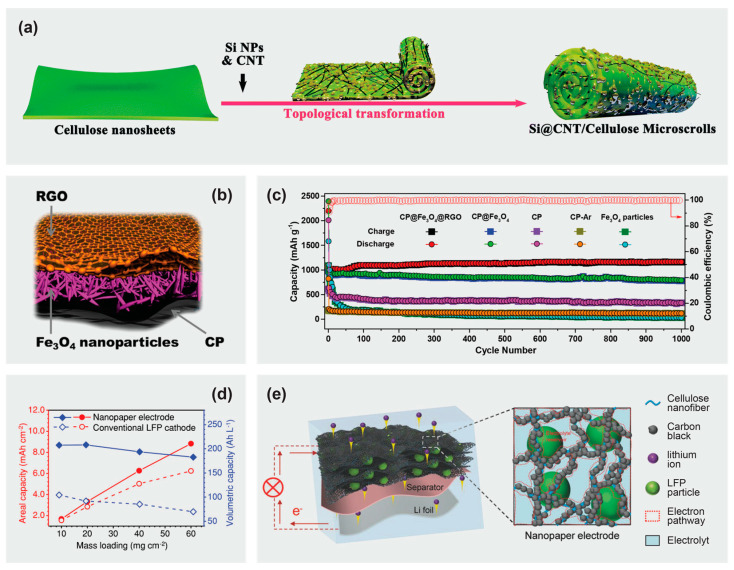
(**a**) Schematic illustration of the deformation process of cellulose nanosheets into Si@CNT/C-microscrolls. Reprinted from [159] with permission from the Royal Society of Chemistry 2020. (**b**) Structure and (**c**) cycling performance of carbon paper@Fe_3_O_4_@reduced graphene oxide anode. Reprinted from [161] with permission from the American Chemical Society 2019. (**d**) Effect of mass loading on the areal and volumetric capacity of the nanopaper electrode vs. conventional lithium iron phosphate electrode. (**e**) Schematic of the electron/ion transportation pathways in the nanopaper electrode. Reprinted from [163] with permission from Wiley-VCH 2018.

### 5.2. Cellulose-Based Polymer Separators

Another important component in a battery is the separator. The separator serves as a solid barrier between the two electrodes to prevent short-circuits and is impregnated with a liquid electrolyte solution that acts as a reservoir of ions for ionic transportation [167,168]. Traditionally, separators are made from polypropylene or polyethylene. While they are low cost and can be produced commercially, such polymers have low porosity and low electrolyte wettability, which affects the performance of the battery. Moreover, polypropylene and polyethylene are non-renewable and not biodegradable. In contrast, cellulose is a sustainable and cost-effective alternative [169]. In this section, separators refer to formed films that are later immersed in a lithium salt and solvent. Cellulose-based separators are usually prepared by electrospinning [170,171], casting [172], and paper making/vacuum filtration [173].

To ensure good electrochemical performance of a cell, characteristics such as thickness [174,175], thermal shrinkage [176,177], porosity [178,179], and mechanical strength [180,181] have to be considered. Lee and coworkers [182] reported an ultra-thin polyethylene-based separator with a self-assembled chiral nematic liquid crystalline cellulose nanocrystal layer (10 µm thickness). During a nano-indentation test, the cellulose nanocrystal layer showed a modulus of >6 GPa, compared to the pristine polyethylene separator of only ~2 GPa. The high mechanical modulus of the cellulose nanocrystal layer was used to account for the lithium dendrite suppression phenomenon observed, where only a smooth and dense layer of lithium was deposited. Moreover, the cellulose nanocrystal/polyethylene separator was able to maintain its coulombic efficiency up to 140 cycles, which is longer than the pristine polyethylene separator that decreased drastically after 80 cycles. When used in conjunction with a 20 µm thin lithium anode, the volumetric energy density of the cell reached 1016 Wh L^−1^ (based on the volume of the electrodes, current collectors and the 10 µm separator).

In order to withstand thermal shrinkage, a heat-resistant, hybrid separator from inorganic xonotlite nanowires with wood pulp fibers was fabricated via a paper making technique [176]. The separator was able to withstand thermal treatment of up to 600 °C without thermal shrinkage. This is especially important as traditional polyolefin separators shrink drastically at high temperatures, resulting in lower porosity and destruction of the separator [183]. Such remarkable thermal resistance was attributed to the inorganic xonotlite nanowires acting as a barrier to the wood pulp fibers, thus protecting them from being distorted [176].

Other than thermal shrinkage, separators with smaller pore size increase the transference number of lithium ions, giving rise to higher battery efficiency. Accordingly, a cellulose acetate-based separator with uniform nanoporous structure was able to promote the uniform transmission of ions. Meanwhile, the cellulose acetate backbone suppressed the transference of anions, and prevented excessive buildup of lithium ions, thus limiting the nucleation and growth of dendrites (Figure 8a,b) [184].

Cellulose not only can play a role as a substrate to combat lithium dendrite growth, but also as a protective layer on a substrate separator. One effective way is to use hydroxypropyl methyl cellulose and SnO_2_ layers which sandwich the polyethylene separator to protect the polyethylene separator from being penetrated by lithium dendrites and also encourage even deposition of lithium. A composite separator was prepared by dip-coating in a hydroxypropyl methyl cellulose solution, followed by the SnO_2_ precursor solution. In this case, the hydroxy group of the cellulose played a pivotal role in ensuring the uniform formation of the SnO_2_ layer and increasing the affinity between the different layers with polyethylene (Figure 8c,d). When compared with other Li || Cu cells using polyethylene and hydroxypropyl methyl cellulose/polyethylene separators which decreased after 50 cycles, the coulombic efficiency of the cell using the composite hydroxypropyl methyl cellulose/SnO_2_ separator was maintained at 99.9% for 170 cycles.

Recently, lithiated cellulose nanocrystals have been incorporated into transient rechargeable batteries as an additive for mechanical reinforcement due to their inherent modulus of 85 J g^−1^ (Young’s modulus to weight ratio) [185]. In this work, cellulose nanocrystals were embedded into a poly(vinyl alcohol) matrix through a non-solvent induced phase separation method. Due to their lyophilic nature, lithiated cellulose nanocrystals improve the electrolyte uptake from 245.7 to 509.1%. The presence of 10 wt% cellulose nanocrystals was also able to reduce the pore size of the poly (vinyl alcohol) membrane from 2−4 µm to 0.5−1.2 µm. With the addition of the biocompatible and biodegradable ionic liquid, the poly(vinyl alcohol)/cellulose nanocrystal-Li separator was able to cycle stably in a transient Li/V_2_O_5_ battery at 55 mA g^−1^ for 200 cycles. When exposed to water, the transient battery disintegrated within 15 min, thus closing the loop in the energy storage field and reducing the amount of hazardous waste left in the environment.

**Figure 8 materials-16-03856-f008:**
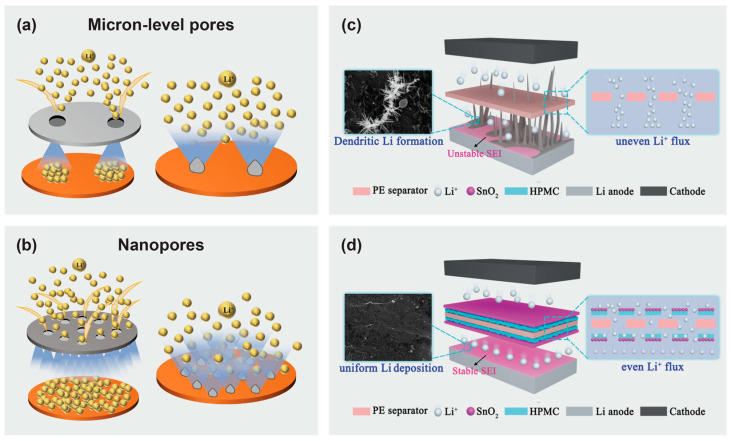
Schematic illustration of the process of Li^+^ transference in two lithium batteries and the influence of different number and size of pores as lithium dendritic nucleation centers on the growth of lithium dendrites based on the (**a**) Celgard 2400 separator and (**b**) cellulose acetate/polyurethane/metal-organic framework separator. Reprinted from [184] with permission from Elsevier 2021. Schematic illustration of Li^+^ deposition process through (**c**) polyethylene and (**d**) SnO_2_/hydroxypropyl methyl cellulose/polyethylene separators. Reprinted from [186] with permission from the American Chemical Society 2022.

### 5.3. Cellulose-Based Polymer Electrolytes

Similar to separators, polymer electrolytes act as a barrier to prevent contact between the positive and negative electrode, thus preventing short circuiting of the battery [187]. Compared to separators, solid polymer electrolytes offer a safer alternative as they are non-leaky and do not contain flammable organic solvents. These all-solid polymer electrolytes usually require a polyethylene oxide segment where the ether oxygen atoms will coordinate with the lithium ions on the solid lithium salt, such as lithium bis(trifluoromethanesulfonyl)imide, to enable ionic transport [188,189]. However, as there is no liquid component at room temperature, the ion transportation kinetics are slower under these circumstances and thus the ionic conductivity usually is at best 10^−5^ S cm^−1^ at room temperature [190,191].

One of the common strategies used to enhance the ambient ionic conductivity, cycling stability, and cycling life is by introducing inorganic fillers. Hu and co-workers [192] used garnet-type Li_7_La_3_Zr_2_O_12_ as a filler in a polyethylene oxide-cellulose electrolyte, where the high shear modulus of the garnet-type Li_7_La_3_Zr_2_O_12_ is said to impede lithium dendrite growth and also prevent the dendrite penetration. Due to its high wettability, the bacteria cellulose template was highly suitable to absorb large amounts of the garnet-type Li_7_La_3_Zr_2_O_12_ precursor solution. With the high aspect ratio of garnet-type Li_7_La_3_Zr_2_O_12_, and the interface between polyethylene oxide and garnet-type Li_7_La_3_Zr_2_O_12_, there were more interconnected channels for lithium-ion transport, leading to a higher ionic conductivity of 1.12 × 10^−4^ S cm^−1^ for the hybrid electrolyte.

Since solid polymer electrolytes usually have lower conductivities, plasticizers and non-flammable ionic liquids are usually introduced to improve ion transportation, resulting in gel polymer electrolytes that contain a liquid phase within the polymer matrix [193]. As there is a liquid phase, the ion transport kinetics are improved and usually ionic conductivities of 10^−4^ to 10^−3^ S cm^−1^ could be achieved at ambient conditions [194,195,196]. This is evidenced by the work of Kale et al. [197], who reported a gel polymer electrolyte based on cellulose triacetate matrix, with poly(polyethylene glycol methacrylate) and the ionic liquid N-methyl-N-butylpyrrolidinium bis(trifluoromethansulfonyl)imide that allows for lithium salt lithium bis(trifluoromethanesulfonyl)imide to move with improved mobility. The room temperature ionic conductivity reached 5.24 × 10^−3^ S cm^−1^, which was explained by the synergistic effect of adding poly(ethylene glycol methacrylate) and cellulose triacetate which contains many polar groups that provide more anchoring sites for lithium ions, thus facilitating lithium-ion conduction.

For polymer electrolytes, mechanical strength, and ionic conductivity usually have an inverse relationship, and this is reflected in gel polymer electrolytes, where the presence of the liquid electrolytes tend to worsen the mechanical performance of the polymer electrolyte [198]. In a similar fashion to solid polymer electrolytes, inorganic fillers such as boron nitride [199], lithium aluminum germanium phosphate [200], and garnet-type Li_7_La_3_Zr_2_O_12_ [201] have been added to gel polymer electrolytes to improve their mechanical performance without compromising ionic conductivity.

### 5.4. Cellulose-Based Applications in Rechargeable Aqueous Batteries

Since most traditional lithium-ion batteries are moisture-sensitive, the presence of water in cellulose poses a potential problem, such as a reduction of battery life cycle [202,203,204]. Conversely, highly water absorbent cellulose is very suitable for aqueous rechargeable batteries, which already have water in their system [205,206,207,208,209,210]. Aqueous batteries are able to function in the presence of water, owing to the usage of novel electrodes that are stable against water [209,211]. Moreover, by utilizing water-based instead of flammable organic electrolytes, aqueous batteries provide a higher level of safety. At the same time, the operating voltage of aqueous batteries is limited by the narrow electrochemical window of water (1.23 V), which lowers the energy density of aqueous batteries [212,213]. To solve this issue, approaches include: (a) Fabrication of novel electrode materials that are stable against water, (b) usage of super concentrated or hybrid electrolytes to suppress water activity [214,215,216], and (c) coatings or membranes that act as a barrier to protect electrode integrity [217]. In this section, we will focus on the roles that cellulose can play in improving and stabilizing the electrochemical performance of aqueous batteries.

One of the ways of fabricating cellulose-based electrodes for aqueous batteries is by forming an aerogel and annealing it at high temperatures. This results in a 3D interconnected structure with a high surface area that improves the electrochemical performance of the battery [218]. Using different sources of cellulose nanofibers and different nitrogen-containing precursors (e.g., urea, hexamethylenetetramine), Li, Jiang and co-workers prepared cathodes from nickel-nickel oxide-cellulose aerogels under hydrothermal conditions at 120 °C [219,220]. The nickel-nickel oxide-cellulose aerogels prepared using hexamethylenetetramine and urea had BET surface areas of 196 and 144 m^2^ g^−1^, respectively. The maximum energy density of the nickel-zinc aqueous battery (441.7 W h kg^−1^ from hexamethylenetetramine vs. 313.4 W h kg^−1^ from urea) benefitted from having a larger surface area and was able to cycle successfully at higher current densities. This enhanced electrochemical performance was attributed to the 3D cellulose framework that allowed for a uniform dispersion of the nickel nanoparticles, thus enabling more active materials to take part in the electron/ion transportation (Figure 9a). Alternatively, instead of turning cellulose into carbonaceous materials via high temperature processing, cellulose can be mixed with other additives to form a free-standing, flexible film.

As a binder, the hydrophilic nature of cellulose is useful in ensuring electrode/electrolyte accessibility [221,222]. Compared to binders such as PTFE and PVDF, cellulose-based binders have been shown to have better electrolyte uptake, thus facilitating ion transport, resulting in improved cycling performance of the cell. As reported by Wang et al., the Zn-MnO_2_ battery with Zn-carbon black-cellulose nanofiber binder anodes was able to achieve higher discharge capacity at 199.3 mAh g^−1^ (87.1% capacity retention, 200 cycles), while anodes prepared with PTFE and PVDF binders could only reach 68.7 and 85.3 mAh g^−1^ [221]. In addition to electrolyte wettability, cellulose can be used in conjunction with other fillers to form protective layers over electrodes. These cellulose-based coatings are able to resist dendrite growth and thus result in a longer battery cycling life [223,224]. Cellulose acetate and cellulose nanofiber-based protective layers were shown to prolong the battery cycling performance of >1000 cycles at 1 A g^−1^, with capacity retention of at least 90%. Such a feat was achievable with thin films of only 40−48 µm in thickness. Interestingly, the combined lignin-containing cellulose nanofiber with a MXene layer only had a maximum stress of 43.7 MPa [224]. This was achievable as cellulose prevented free water molecules from reaching the Zn anode by trapping them, thus preventing corrosion of the Zn anode [224,225]. Moreover, the plate-like morphology of MXene has a similar lattice parameter to the zinc anode at (002) plane, which encourages epitaxial dendrite growth laterally along the zinc anode [224].

As a binder, cellulose improves the mechanical stability of the electrode and in some cases, imparts flexibility upon the electrodes [226]. As exemplified by Wang and co-workers, they prepared a free-standing composite anode film made from Zn microspheres, carbon nanotubes, and nanocellulose. This single matrix-integrated system was able to function in spite of the elimination of current collectors and binders through vacuum filtrating the anode, cellulose fiber separator, and cathode consecutively. The inclusion of cellulose as a binder or component in all layers resulted in a thin film of ~395 µm with good interfacial contact, with a discharge capability of 146.8 mAh g^−1^ at 1 A g^−1^ [227].

**Figure 9 materials-16-03856-f009:**
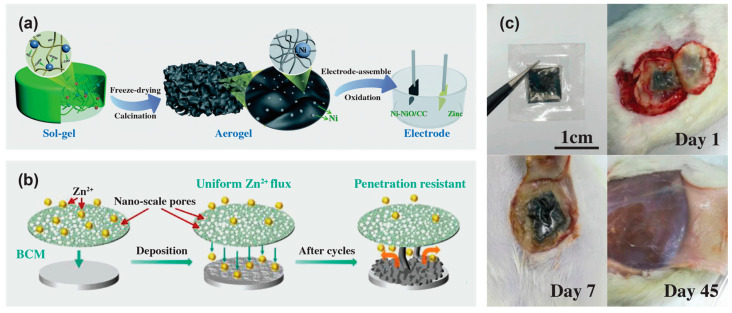
(**a**) Schematic illustration of the process of the nickel-nickel oxide-cellulose composite electrode. Reprinted from [220] with permission from the Royal Society of Chemistry 2020. (**b**) Schematic illustration of working mechanism of bamboo cellulose membrane separator. Reprinted from [228] with permission from the Elsevier 2022. (**c**) Optical images of zinc thin film battery before implantation, in vivo degradation of zinc thin film battery inside the subcutaneous tissues of rats on days 1, 7, and 45. Reprinted from [229] with permission from the Elsevier 2022.

Due to the presence of water, aqueous batteries also need to account for the hydrogen evolution reaction (HER) that heavily impacts local pH values at the interface between the electrolyte and electrodes. Hence, the separator must be able to resist chemical changes in addition to having good mechanical strength [230]. Fu and co-workers prepared a bamboo-derived cellulose membrane with tensile strength of up to 81 MPa which was able to cycle successfully in electrolytes at pH 3.6 and pH 13.1 (Figure 9b) [228]. In the Zn||Zn symmetric cell containing the cellulose membrane, it was able to cycle for >5000 h at 0.1 mA cm^−2^ 0.5 h in 3 M Zn(CF_3_SO_3_)_2_ and 1 M KOH. This was notably higher than the commonly used glass fiber separator, which could only cycle for 30 and 4 h in acidic and alkaline conditions, respectively. Different strategies to combat zinc dendrites also include combining zincophobic cellulose-based separators with glass fiber separators [231], or a hydrophilic cellulose membrane modified with zirconium oxide particles [232] and hierarchically porous persimmon branch carbon with carbon dots [233]. They work by ensuring uniform, lateral zinc deposition on the surface of zinc anode, resulting in a longer cycling life.

Other than imbuing the material with better mechanical properties, the inclusion of cellulose as a natural polymer can also be used to produce biocompatible and biodegradable aqueous batteries [234]. Recently, Kong and co-workers prepared a cellulose aerogel-gelatin solid electrolyte with ionic conductivity of 1.23 × 10^−2^ S cm^−1^ that could withstand a high extent of bending at 120° and is able to recover completely from compression, whereas a control gelatin electrolyte was entirely damaged [229]. They further encapsulated the cellulose aerogel-gelatin solid electrolyte with the electrodes in a silk packaging and implanted the battery in the subcutaneous region of rats. After 30 days, the battery was fully broken down and the rats survived without any disorders or disability. After 45 days, the injury of the rats started recovering back to normal (Figure 9c). This displayed the potential of the cellulose aerogel-gelatin solid electrolytes as a biocompatible, biodegradable battery that can be degraded once triggered by the environment. Moreover, cellulose hydrogels can be used to encapsulate super concentrated salts or antifreeze agents. This would enable the batteries to work in low temperature conditions [235,236].

## 6. Cellulose-Based Composites for Water Splitting Catalysis

With the increasing intensity of the global energy crisis, substantial efforts have been made to reduce carbon emissions by switching to renewable energy sources. Hydrogen is considered as the most promising energy source to replace fossil fuels in the future [237] due to its naturally high energy density (with respect to other common fuels). When combining hydrogen with oxygen in a fuel cell, the hydrogen can be converted into electrical energy and water formed as the byproduct. Generally, hydrogen can be generated from fossil fuels or biomass, or it can be generated via water splitting catalysis, by breaking water into H_2_ and O_2_. Work on water splitting focuses primarily on electrocatalytic [238], photo-catalytic [239], and photo-electrocatalytic [240] systems. With regard to electrocatalysis, cellulose-based materials present great potential as a substrate for the growth of the active catalyst layer because of the porous microstructure and large amount of hydroxyl and epoxy functional groups over the large surface area of the cellulose fibrils. These functional groups are known to have reactive surfaces that can facilitate strong binding and better mass-transport. Moreover, flexibility of the substrate can reduce internal strain during electrochemical cycling.

### 6.1. Cellulose-Based Photocatalysts

The photocatalytic process decomposes water into its constituents (H_2_ and O_2_) using a catalyst and natural light. This process was first discovered by Fujishima and Honda who realized that TiO_2_ can be used in water splitting with light assistance. Since then, TiO_2_ has been widely used as the benchmark in photocatalysis [241]. To enable large-scale production of hydrogen through photocatalytic water splitting, irradiation on a large catalyst area is crucial. By emulating solar cells, large area TiO_2_ photocatalytic films were fabricated from a cellulose/catalyst suspension [242]. However, despite the films being photocatalytically active, their durability during reaction needed further improvement because of the swelling of cellulose and the breaking of the cellulose layer while releasing the hydrogen gas.

Cellulose nanofibers can serve as an ideal template for producing functional, fibrous 3D nanostructures with porous structures and high surface area, thus offering excellent charge transport properties and long optical paths for efficient light absorption. For instance, a 3D fibrous TiO_2_ nanotube architecture was fabricated as a photoanode by the atomic layer deposition of TiO_2_ films over a cellulose nanofiber template [243]. Taking advantage of the inherent hydrophilic nature of the cellulose, a unique out-of-electrolyte capillary photoelectrochemical setup was fabricated to conduct a water splitting reaction using capillary force to supply the electrolyte to the active area. It was observed that higher reaction kinetics and higher efficiency can be accomplished from the capillary photoelectrochemical design, in comparison with the conventional in-electrolyte reaction.

Similarly, a 3D TiO_2_ fiber-nanorod for photoelectrochemical water splitting was grown on a ZnO-coated cellulose nanofiber template using atomic layer deposition [244]. Interaction between the ZnO layer and TiCl_4_ precursor vapor produced polycrystalline TiO_2_ thin films, seeding the growth of TiO_2_ nanorods via the Kirkendall effect. The high density nanorod branches that grew into mesoporous cellulose nanofiber networks exhibited large porosities which further enlarged the surface area for photoelectrochemical water splitting. As a result, the 3D TiO_2_ fiber-nanorod heterostructures showed higher photocurrent and photoelectrochemical efficiency relative to TiO_2_-ZnO bilayer tubular nanofibers and TiO_2_ nanotube networks.

Another successful example of paper-based electrodes was demonstrated by the in situ crystallization of a carbon nitride precursor on conductive carbon paper substrate, followed by thermal condensation [245]. The carbon paper substrate can be easily prepared by the pyrolysis of a filter paper, and the shape and structural integrity were retained even after carbonization at high temperature. The resultant hierarchical carbon nitride/carbon paper hybrid structure revealed the homogeneous growth of vertically aligned carbon nitride microstructures from the carbon paper, which yielded enhanced crystallinity and close contact with the carbon paper. Working as a photoanode in a water splitting photoelectrochemical cell, the freestanding electrode showed high electrical conductivity and good photoelectrochemical performance.

### 6.2. Cellulose-Based Electrocatalysts

Electrocatalytic water splitting is the process by which water is broken down into gaseous hydrogen and oxygen by applying electrical energy. Efficient water splitting requires electrocatalysts to elevate the half-cell reactions, which are the oxygen evolution reaction (OER) and the hydrogen evolution reaction (HER) [238,246]. As cellulose substrates are nonconductive, several techniques have been developed to impart conductivity to them, such as printing [247], simple solvent evaporation [248], electronic sputtering [249], electroless plating [250], etc.

Bhattacharyya et al. [251] fabricated a conductive and flexible paper-electrode by first transforming an insulating substrate into a conductive current collector via electroless metal plating of metallic nickel nanoparticles on cellulose paper, followed by modification of the nickel-paper with nickel-iron oxyhydroxide and nickel-molybdenum bimetallic alloys (Figure 10). The catalytic paper-electrodes were found to require overpotentials of 240 and 32 mV at 50 and −10 mA cm^−2^ to drive OER and HER, respectively. In addition to good water splitting activities, the flexible paper-electrodes exhibited good mechanical robustness under harsh alkaline conditions due to the favorable electrode configuration and porous hierarchy that benefited close electrical connections and enhanced mass and charge transport.

Likewise, a catalytic paper-electrode comprised of Ni-P-B could be fabricated by depositing conductive catalysts on a paper substrate that was pre-treated with nickel boride nanoparticles using the electroless plating approach [250]. The resultant Ni-P-B/paper electrode displayed low overpotentials of 76 mV and 263 mV for the HER and OER, respectively, in alkaline conditions. Moreover, the paper electrode demonstrated high stability which could survive at a large current density of 1000 mA cm^–2^ for more than 240 h with negligible performance degradation. When compared to a Ni-foil substrate, the electrochemical performance of the paper-electrode was superior, due to the unique porous structure and coarse surface morphology of paper which provided a large electrochemically active surface area and enabled nano-electrocatalyst formation, instead of the formation of a dense catalyst layer on the Ni-foil substrate.

When cellulose-derived bamboo fiber was adopted for the fabrication of a free-standing and flexible current collector, the insulating cellulosic bamboo fiber was first transformed into a conductive substrate by the electroless deposition of Co-P, followed by direct electrodeposition of cobalt-iron (CoFe/Co-BF) and cobalt-molybdenum (CoMo/Co-BF) nanostructures for OER and HER applications, respectively (Figure 11a) [252]. The resultant bimetallic CoFe/Co-BF anode and CoMo/Co-BF cathode demanded overpotentials of 250 mV at 50 mA cm^–2^ and 46 mV at 10 mA cm^–2^, respectively, in alkaline electrolyte, which were lower than other Co-based electrocatalysts (Figure 11b,c). When operated in proto-type alkaline electrolyzer, the electrocatalysts developed required a low bias voltage of 1.55 V to achieve a current density of 20 mA cm^–2^ that persisted for 24 h. The proposed system with a cellulose-based bamboo fiber substrate has a macro-porous network of hexa-filament microfibrils which favor electrolyte pathways, giving rise to greater reaction kinetics, thereby demonstrating superior electrocatalytic activities.

Apart from being a flexible electrochemical substrate, cellulose-based materials have been used to substitute synthetic polymers (e.g., Nafion) for immobilizing powder catalysts. In this study, a flexible HER electrode comprised of nitrogen-doped molybdenum carbide nanobelts, graphene nanosheets, and nanocellulose was prepared via a vacuum filtration approach [253]. Compared to a Nafion-bonded electrode which possessed an overpotential of 180 mV at 10 mA cm^−2^, the nanocellulose bonded electrode displayed a lower overpotential of 163 mV at 10 mA cm^−2^ in an acidic medium, due to the good bonding between cellulose nanowhiskers and nitrogen-doped molybdenum carbide nanobelts.

Cellulose fibers can work as a rich carbon template when transforming cellulose-rich waste-paper into electrodes via a carbonization process. This was demonstrated by Mu and co-workers [254] who used cobalt as a self-promoter catalyst to induce the structural transformation of a paper’s fibers into self-standing heteroatom co-functionalized carbon nanorods, for use as a bifunctional electrocatalyst. The resultant electrocatalyst is primarily comprised of a Co_9_S_8_ and Co-N-integrated carbon nanorod framework (Co_9_S_8_@Co-N/C). Benefiting from this synergistic chemical composition, the bifunctional electrocatalyst delivered 10 mA cm^–2^ at a cell voltage of only 1.61 V and exhibited superior durability of more than 70 h with negligible degradation. From the viewpoint of recycling, this study has demonstrated a sustainable solution to turn paper waste into value-added products for emerging applications.

Taking advantage of 3D network structure and outstanding mechanical properties of native cellulose-I nanofibers, bacterial cellulose structures can be employed to create flexible heteroatom-doped carbon nanofiber aerogels as water splitting electrocatalysts [255]. Wu et al. [256] fabricated a Mo_2_C-based HER electrocatalyst, comprising Mo_2_C nanoparticles embedded in bacterial cellulose derived N-doped carbon nanofibers via a solid-state reaction. Due to the combined effect of Mo_2_C nanoparticles and N-doped carbon nanofibers, the resultant HER electrocatalyst exhibited 10 mA cm^–2^ at an overpotential of 167 mV and an exchange current density of 4.73 × 10^−2^ mA cm^−2^ in acidic solution. It was further demonstrated that the developed Mo_2_C-based HER electrocatalyst exhibited electrocatalytic activity in a broad pH range (pH 0–14), which is highly desirable to achieve overall electrochemical water splitting since most of the non-noble metals based OER electrocatalysts work well only in basic or neutral media, while the majority of the non-noble-metals-based HER electrocatalysts perform well only in acidic electrolyte.

## 7. Conclusions and Future Outlook

In this review, we have presented the recent developments in the preparation of cellulose-based conductive composites, and their potential applications in energy-related functions. Unlike the conventional application of cellulose in biofuel generation, these applications rely on the polymeric nature of cellulose. As a naturally occurring polymer commonly found in plants, fungi, and algae, cellulose possesses many unique features. It is fully biocompatible, whilst displaying extraordinary optical, mechanical, and thermal properties that are rarely found in other polymers. The presence of carboxylic acid and hydroxyl functional groups along the cellulose chain allows further functional group modifications, which in turn enables customization of the polymer for specific applications. When combined with other components such as metal or semiconductor nanoparticles, carbon-based derivatives, organic polymers, or metal-organic frameworks, the composites formed exhibit many new and interesting properties.

Various strategies have been developed for constructing advanced cellulose-based composites with desirable structures and enhanced electrochemical performances for use in flexible energy-related devices. Notably, cellulose-based composites with superior mechanical strength and low density play a vital role in flexible energy harvesting devices, such as solar cells, and triboelectric, piezoelectric, and thermoelectric generators. Utilizing the high thermal and structural stability of nanocellulose, together with its wettability, composites with good hydrophilicity and porosity can be created. Such cellulose-based composites also demonstrate great potential as flexible electrolytes and separators for energy storage applications. Importantly, cellulose with high specific surface area provides abundant sites for electrochemically active materials to be anchored on the surface, which can enhance the electrochemical performance of the catalytic electrodes in water splitting. Compared with conventional solid devices, these cellulose-based composites are far more renewable and flexible, whilst maintaining excellent performance.

Despite the current achievements in the field, the use of cellulosic materials in sustainable energy-related devices is still challenging, primarily due to the laborious nature of converting bulk cellulose to its nanoscale derivatives. Scaling up the production of appropriate cellulose derivatives to industrial levels can be extremely challenging, as cellulose fibers have a complex hierarchical structure, comprising microfibrils and crystalline regions that make it difficult to process into a uniform composite material. The complex processing of cellulose requires specialized equipment and techniques to achieve consistent product quality, which can add to the cost and complexity of composite production, thereby impeding their real-life applications. The advantages and disadvantages of cellulose in different energy devices are summarized in Table 1.

Cellulose has been used as a component for many electrical insulating materials. However, the intrinsically insulating nature of cellulose is unfavorable for the electrochemical performance of the derived energy-related devices. Additionally, the abundance of hydrogen bonding sites that benefit the high performance of cellulose-based composites can also cause the material to have high affinity for moisture/water in the environment. Commonly, this results in mechanical failure and accelerated degradation of the composites, limiting the durability of the material. The hydrophilic nature of cellulose also limits its compatibility with different kinds of polymer matrices—particularly those which are hydrophobic or have low surface tension. While cellulose has many desirable and advantageous properties, its simple inclusion into composites results in materials which may not always meet the performance requirements for certain energy applications or functional devices. Further research and development are required to improve their properties, useability, and functionality.

Currently, the integration of cellulose-based materials with existing technologies in energy applications and functional devices requires the development of new production processes. In order to facilitate the large-scale adoption of cellulose-based composites, future research and development should be directed to the fundamental scientific understanding of the processing of nanocellulose from the bulk, as well as advanced fabrication techniques and characterization for cellulose functionalization and modification. Extended efforts focusing on the rational design of the constituent materials would be welcomed, as well as surface/interface engineering to compensate for the lack of conductivity of nanocellulose and to improve the water resistance for enhanced durability. Based on these strategies, well-designed cellulosic materials will serve as an important and multifunctional platform for future energy applications and other functional devices.

## Figures and Tables

**Figure 1 materials-16-03856-f001:**
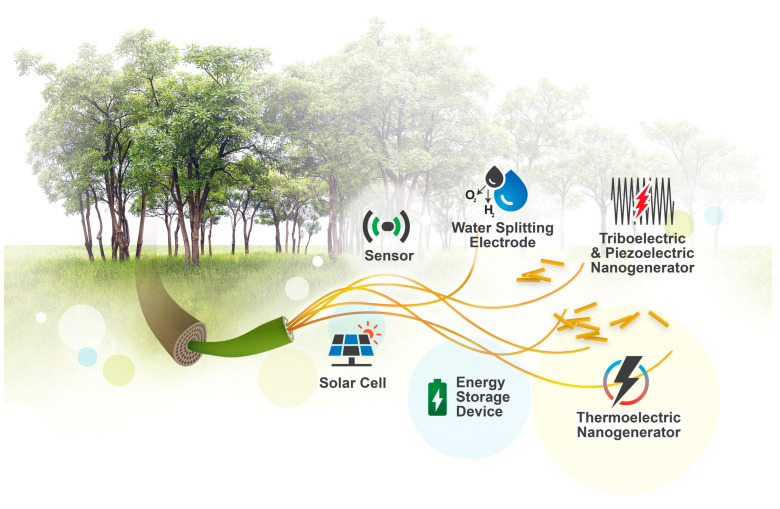
The development of cellulose-based composites in the advancing field of flexible wearable electronics, thermoelectric nanogenerators, mechanical energy nanogenerators, sensors, electrodes, and photovoltaic solar cells.

**Figure 2 materials-16-03856-f002:**
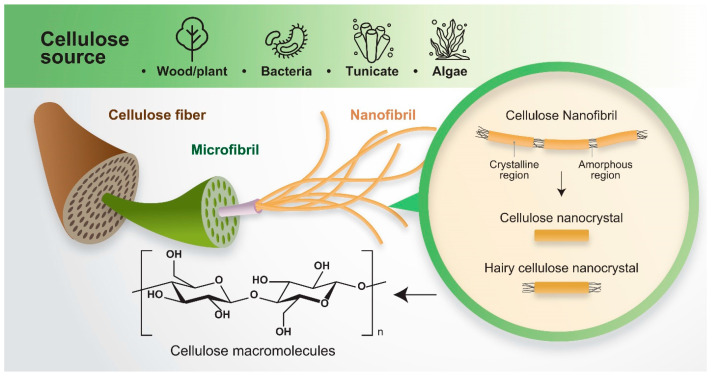
Schematic illustration of hierarchical structure of cellulose from cellulose sources to nanocelluloses types and chemical structure.

**Figure 3 materials-16-03856-f003:**
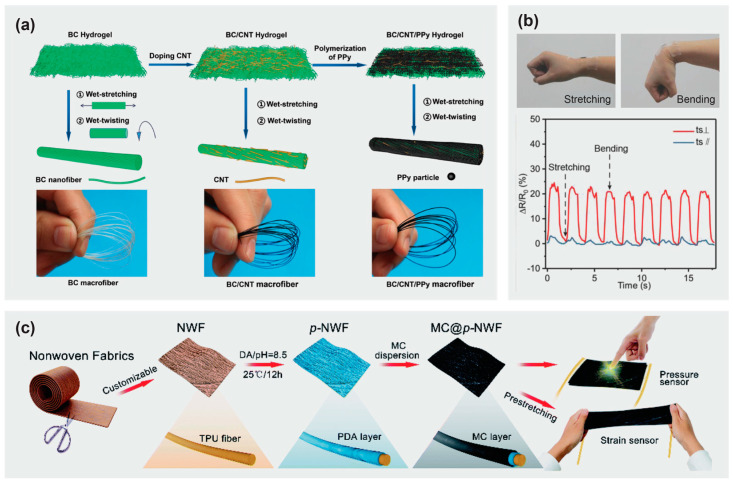
(**a**) Schematic illustration of BC, BC/CNT, BC/CNT/PPy macrofibers preparation process. Reprinted from [58] with permission from Springer 2022. (**b**) Carbonized crepe paper strain sensor on a human wrist and relative resistance change of strain sensor in cyclic stretching-bending of the wrist. Reprinted from [59] with permission from Wiley-VCH 2018. (**c**) Schematic illustration of the fabrication process of wearable strain/pressure sensor based on MXene/cellulose nanocrystal coated nonwoven fabrics. Reprinted from [60] with permission from the Royal Society of Chemistry 2020.

**Figure 10 materials-16-03856-f010:**
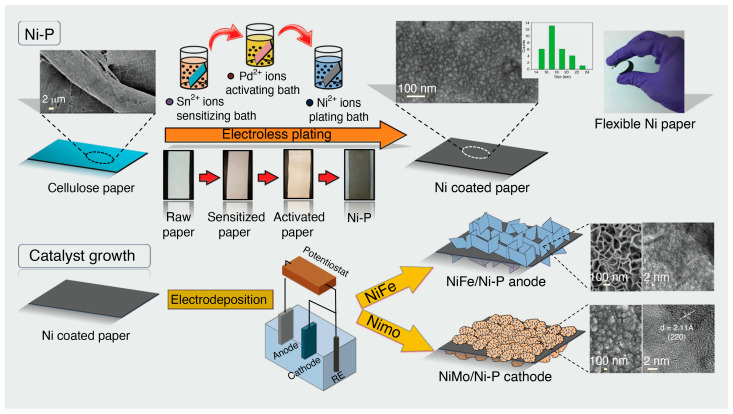
Schematic illustration of the fabrication process of active substrates from paper as electrodes for HER and OER applications. Reprinted from [251] with permission from the Springer Nature 2018.

**Figure 11 materials-16-03856-f011:**
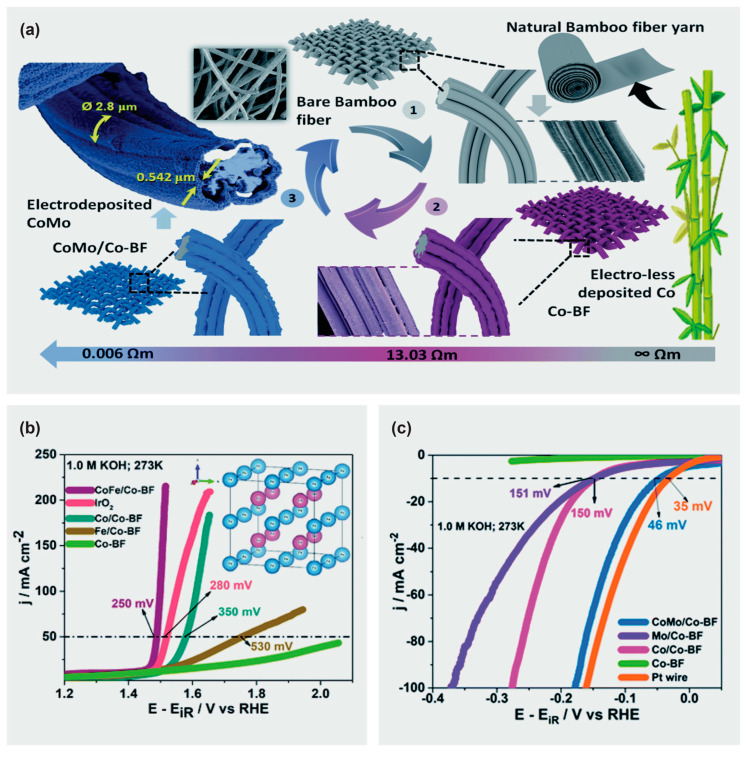
(**a**) Schematic illustration of the fabrication process of active substrates from bamboo fibers as electrodes for HER and OER applications. Electrocatalytic performances of (**b**) CoFe/Co–BF anode, and (**c**) CoMo/Co–BF cathode in alkaline electrolyte. Reprinted from [252] with permission from the Royal Society of Chemistry 2021.

**Table 1 materials-16-03856-t001:** Advantages and disadvantages of cellulose in energy devices.

Energy Devices	Advantages	Disadvantages
Flexible and Wearable Electronics	Highly flexible for integration into wearable devices, improving user comfort and ease of wear High level of transparency and suitable for use in transparent and translucent devices Biocompatible in nature that is not harmful to the environment	Proper encapsulation is needed due to intrinsic hydroscopic nature of cellulose in which the structure integrity degrades over prolonged period Chemical modification or doping is needed due to intrinsic low conductivity which limits its use in electronic devices that require high conductivity
Piezoelectric nanogenerators (PENGs)	Biocompatible and non-toxic, suitable for powering implantable medical devices in biomedical applications Relatively high piezoelectric coefficient, whereby a large amount of electrical energy can be generated when subjected to mechanical stress or vibration	Highly sensitive to moisture which can affect mechanical properties and piezoelectric performance Low output voltage which may limit its use in certain applications where high output voltage is required
Triboelectric nanogenerators (TENGs)	Relatively high dielectric constant which means it can store a large amount of electrical charge Highly flexible and elastic that can be used to fabricate wearable TENGs such as clothing and wearable devices, enabling energy harvesting from the movement of wearer	Highly sensitive to moisture which can lead to electrical short-circuits, reducing efficiency of the TENGs Low output voltage and may limit its use in certain applications where high output voltage is required
Thermoelectric generators	Overcome the brittleness of inorganic semiconductors to accomplish flexible thermoelectric generators	Limited to low-temperature applications and may not be able to withstand high-temperature thermoelectric applications
Solar cells	Highly transparent to visible light which makes it a suitable material for use as a substrate in transparent solar cell Low cost and attractive option for mass production	Limited light absorption properties which can limit its use as active material in solar cell Highly sensitive to moisture which can cause degradation of the material and lead to a reduction in efficiency of the solar cell
Batteries	High versatility that can be transformed into different components of a battery, from the electrodes to the separator and binder. High porosity which allows for the diffusion of ions and electrolytes in battery, leading to improved battery performance Low cost and attractive option for mass production	Highly sensitive to moisture, which may remain as a contamination in non-aqueous batteries and reduces their lifetime and capacity Limited thermal stability which can limit its use in high temperature applications. Limited chemical stability which can undergo chemical reactions with certain electrolytes and other battery components, leading to degradation of material and reduced battery performance
Water splitting catalysts	High porosity which allows for the diffusion of reactants and products in water splitting process High surface area which can increase the efficiency of water splitting reaction and improve the performance of the process	Low electrical conductivity which can limit its efficiency as an electrode material in the water splitting process Limited light absorption properties which can limit its use as photoelectrode material in the water splitting process Limited stability when use in acidic or basic solution

## Data Availability

No new data were created or analyzed in this study. Data sharing is not applicable to this article.

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
