# Peer review of "Advances in Cellulose-Based Composites for Energy Applications"

_materials, 2023, doi:10.3390/ma16103856_

Round 1

Reviewer 1 Report

In this manuscript, the authors reviewed the recent developments of cellulose-based materials for energy application, underlying challenges, and outlook in these fields. This work is interesting and it will help to get familiar with cellulose-based materials for energy applications. It can attract researchers in related fields. A major revision is recommended and it can be accepted after addressing the following comments.

1. In the part of introduction, Line 47~Line 68, bacteria cellulose and algae cellulose have been mentioned in the first paragraph of introduction, these two paragraphs could be integrated into the first paragraph.

2. Cellulose is a versatile biopolymer with many types, including native cellulose, cellulose nanofibril, cellulose nanocrystal, bacteria cellulose, and various cellulose derivatives. Despite the diverse types, each exhibits unique properties that can be used for specific applications. Unfortunately, the relationship between cellulose types and their corresponding applications was not thoroughly explored in the part of introduction.

3. In part 2. Classification of Cellulose and Processing, it’s better to introduce the cellulose types according to the following:

2.1 cellulose nanocrystals (CNCs),

2.2 cellulose nanofibrils (CNF),

2.3 amorphous nanocellulose,

2.4 hairy cellulose nanocrystals.

4. In line 199 ~ Line 206, there is no numerical value of reference samples. Eg. Line 199 high conductivity, what’s the conductivity of hybrid film? Please provide the numerical value.

5. In section 3.2. Cellulose-based Materials in Flexible Sensors, the author listed four types of materials, it is recommended to cite the following literature:

(1) A multiscale biomimetic strategy to design strong, tough hydrogels by tuning the self-assembly behavior of cellulose.

(2) Cellulose-based flexible and wearable sensors for health monitoring

6. In section 4.3. Solar Energy Conversion, Line 642~Line 645. Apart from the treatment of cellulose surfaces like coating, are there any other solutions to solve this problem?

7. In section 5.1. Cellulose-based electrodes, there is an unnecessary spacing in the title number. Additionally, besides anode materials used in Li-ion batteries, could cellulose raw materials also be utilized in the production of anode materials in batteries such as Zinc-ion batteries, Na-ion batteries, or any other types?

8. In the last part of this review, although the crystalline structure of cellulose was introduced in the introduction section, there are scarcely any examples of using the cellulose crystalline structure to design materials. It’s better to be appropriately elaborated within the outlook.

Extensive editing of the English language is required.

Reviewer 2 Report

This review provided specific examples of cellulose-based materials and their applications as substrates in the energy storage field, as well as their benefits over traditional plastic and metal substrates. The authors have brought examples to illustrate the potential impact of these materials on the environment and the economy and inspire more interest in their use. They highlighted most of the advantages of cellulose properties. I would suggest they consider discussing the challenges and limitations of using cellulose-based materials, as well as potential solutions to these issues in section 7 (Conclusion and Future Outlook). This will help readers to better understand the feasibility and practicality of using these materials in various applications and will provide a more balanced perspective on their potential benefits and drawbacks.Regarding the structure of the text, the sections were well divided and it is easy to flow the reading. 

There are small English mistakes, but English is my second language and I do not feel able to correct them. 

However, these mistakes do not reduce the readability of the text.

Reviewer 3 Report

This is an interesting and comprehensive review about the development of cellulose-based materials for energy-related applications. Various applications are discussed e.g. solar applications, thermoelectric applications, sensors etc. The paper is well organized. For example, in every subsection e.g. for triboelectric applications or for batteries there is a brief paragraph to introduce the reader to the basic principle of each application. This is very useful since the review covers a broad range of applications and a reader may not be familiar with all of them. The size of the subsections is balanced and the text is informative and a large number of articles are discussed.

 I suggest minor revision. Below are some issues to be addressed.

1.      In general, the paper is well written, however, the biodegradability and abundance of cellulose and the need for sustainable materials etc. are repeatedly mentioned throughout the text too many times e.g. lines 230-231, lines 183-184, lines 450-41 etc.

2.      In the first 6 pages there is only one Figure. I suggest to replace Figure 1 with another one showing the chemical structure of cellulose, the arrangement of chains in each type or the in most common type I. The current Figure 1 could be embedded in the new Figure 1 as subfigure 1a.

Reviewer 4 Report

The manuscript reviews the potential applications of cellulose based composites for energy applications. The paper is very well written and will be a very useful contribution to cellulose based energy materials. The authors have well described the unique properties of cellulose which make cellulose based materials suitable for certain applications. I would like to recommend the paper for publication. I would also suggest only a minor addition of a table which summarizes the different applications of cellulose in energy devices and the advantages and disadvantages. I also feel a brief discussion on the current status of these promising applications of cellulose based materials will be useful. In particular, a discussion on what is preventing the real life application. The brief mention of hairy cellulose catches my attention and I would be interested in more of its applications. 

Round 2

Reviewer 1 Report

It can be accepted for publication. 

Certain modification.